evolution, genetics, ecology

gene expression evolution, complex traits, early burst, key innovation, snake venom

**Author for correspondence:**
Agneesh Barua
e-mail: agneesh.barua@oist.jp

# Toxin expression in snake venom evolves rapidly with constant shifts in evolutionary rates

Agneesh Barua[1] and Alexander S. Mikheyev[1,2]

[1]Ecology and Evolution Unit, Okinawa Institute of Science and Technology Graduate University, 1919-1 Tancha, Onna-son, Kunigami-gun, Okinawa-ken 904-0495, Japan
[2]Evolutionary genomics group, Australian National University, Canberra ACT 0200, Australia

AB, 0000-0002-8347-2171

Key innovations provide ecological opportunity by enabling access to new resources, colonization of new environments, and are associated with adaptive radiation. The most well-known pattern associated with adaptive radiation is an early burst of phenotypic diversification. Venoms facilitate prey capture and are widely believed to be key innovations leading to adaptive radiation. However, few studies have estimated their evolutionary rate dynamics. Here, we test for patterns of adaptive evolution in venom gene expression data from 52 venomous snake species. By identifying shifts in tempo and mode of evolution along with models of phenotypic evolution, we show that snake venom exhibits the macroevolutionary dynamics expected of key innovations. Namely, all toxin families undergo shifts in their rates of evolution, likely in response to changes in adaptive optima. Furthermore, we show that rapid-pulsed evolution modelled as a Lévy process better fits snake venom evolution than conventional early burst or Ornstein–Uhlenbeck models. While our results support the idea of snake venom being a key innovation, the innovation of venom chemistry lacks clear mechanisms that would lead to reproductive isolation and thus adaptive radiation. Therefore, the extent to which venom directly influences the diversification process is still a matter of contention.

## 1. Introduction

Key innovations are adaptations that provide an ecological opportunity by enabling the utilization of previously unexplored niches [1–3]. This enables animals to colonize new environments and in turn facilitates ecological speciation [4,5]. While this concept has considerable intuitive appeal, the idea of key innovations is not exempt from ambiguity and controversy. Throughout history, key innovations have been defined in numerous ways [1]. The many definitions often lead to confusion regarding what key innovations are, and the expected patterns they should exhibit. The long-standing belief is that key innovations lead to the spread of ecological adaptive zones whose eventual outcome is species diversification or adaptive radiation [3,4,6–8]. However, as reviewed by Rabosky [9], key innovations should not be considered the sole reason for differential rates of species diversification. Rather, the role of key innovations should be focused on providing entry into novel ecological niches or adaptive zones, and studies should aim to identify specific shifts in tempo and mode of phenotypic evolution of the assumed key innovation [9,10]. Ecological speciation, i.e. speciation driven by differences in ecology, is considered the primary mode by which adaptive radiation can take place, and as various traits produce specific differences in ecology, certain traits are more strongly associated with the radiation process than others [3,11]. Therefore, it is vital

to explain the effect of trait differences and how they contribute to species diversification, overall phenotypic disparity and ecological divergence.

Evolutionary models are extensively used to study trait evolution and have been used to model everything from body shape evolution to gene expression level evolution [12,13]. Therefore, it is not surprising that evolutionary models are also widely used to study key innovation. However, rarely does one model consistently explain the evolution of key innovations (or traits believed to be key innovations). Some traits are better explained by Brownian motion (BM), others by Ornstein–Uhlenbeck (OU) models; some traits fit a single-peak OU model better, while others a multi-peak OU model; other traits fit neither BM nor OU processes well [14–18]. Along with BM and OU models, it is also possible to model early burst (EB). An EB in speciation rate and trait evolution is believed to be the predominant pattern in adaptive radiation [11,19,20]. While it is possible to model EB, evidence for it is rarely observed in comparative data [21]. The often-conflicting results between these models warrant cautious interpretation of features like evolutionary rates [14]. Perhaps one limitation of these models is using a Gaussian process to model continuous trait evolution. Evolutionary processes can result in changes that are too abrupt to be accounted for by a Gaussian process [22]. Pulsed models, however, can account for abrupt shifts in the continuous character evolution that conventional evolutionary models cannot easily explain [22]. For example, using this approach, Landis & Schraiber found that body size evolution is better represented by rare stochastic pulses of diversification than by conventional EB or multi-optima OU models [11,20]. Therefore, examining traits using a pulsed model of evolution might reveal previously unresolved evolutionary trends.

The complex nature of traits makes it difficult to ascertain how individual components of the complex phenotype contribute towards evolutionary innovation. It also makes it difficult to discern the specific evolutionary trajectories experienced by individual genes. Since gene expression represents the contribution of an individual gene, especially in highly specialized tissues, it is ideal for identifying gene-specific trends in evolutionary rates. This modular nature makes gene expression in certain tissues highly autonomous, such that the activities of genes within that system depend very little on elements outside of it, facilitating the production of specific heritable variations and evolutionary innovations [23,24]. Highly tissue-specific genes would also likely reduce significant pleiotropic constraints and cross-phenotype associations, helping to discern the unique trajectories experienced by individual genes [25]. Despite the usefulness of modelling gene expression, relatively few phenotypes can be meaningfully reduced purely to gene expression levels, making the study of gene expression variation in phenotypic evolution difficult.

Exceptionally, snake venom, which is a complex phenotype composed of secreted proteinaceous mixtures, can essentially be reduced to expression levels of each of its constituent components. This enables us to understand the contribution of each of the constituent genes towards phenotypic variation. Venom toxins can have both agonistic and antagonistic interactions with other toxin components, but how they influence other traits outside the venom system is unclear. On one hand, venoms are integrated systems with different toxins acting in concert to immobilize prey [26]. On the other hand, whether this mode of action introduces an evolutionary constraint is less clear, since there is little phylogenetic covariance between components, and gene–environment constraints appear to act on individual loci, independent of co-expression patterns between toxin genes [27,28].

Each component of the snake venom cocktail is a toxin that can be quantified and traced to a distinct genomic locus [29–32]. Changes in expression levels of individual toxins alter their abundance in the venom, thereby influencing venom efficacy [32–34]. This alteration in venom efficacy impacts the feeding ecology of snakes, which in turn determines how snakes adapt and colonize new niches [35,36]. The strong ecological and evolutionary consequence of toxin expression variation allows us to characterize the gene expression levels of toxins as polygenic phenotypes and trace venom evolution over macroevolutionary timescales.

The idea that venom is a key innovation and that it underlies the extensive radiation of snakes is pervasive in the literature [26,37–41]. Yet, few studies have examined long-term changes in evolutionary rates of venom gene expression in snakes. There are numerous studies that have examined the role of venom in lineage diversification in other taxa [42–45]. In blenny fish, the presence of a venom system in the form of a buccal gland and fang is associated with higher rates of diversification [44]. In tetrapods, the evolution of venoms and poisons is typically associated with an increase in diversification rates (except in amphibians) [42]. There is also a substantial amount of literature suggesting the role of diet in lineage diversification [45–48]. Since snakes use venom primarily for prey procurement, alterations in venom and diet could have an effect on diversification in venomous snakes.

Key innovations, however, have more features than just causing lineage diversification. Key innovations contribute to the expansion of ecological ranges, represent optimal adaptations, and usually undergo changes in evolutionary rates to fill morphospace [49]. Restricting the role of key innovations to just diversification ignores these features and removes focus from evolution of the key innovation itself [9,49]. In this study, we specifically focus on the evolution of snake venom. We use a comparative dataset of snake venom gene expression to identify shifts in phenotypic macroevolutionary rates, which are characteristic of key innovations [9]. To further characterize the patterns of venom evolution, we estimated long-term changes in evolutionary rates of venom gene expression and also fitted the data to several trait evolution models. Our results revealed that toxin expression in snake venom evolves very rapidly and has experienced numerous shifts in evolutionary rates over the past 60 million years.

# 2. Results

## (a) Phylogeny and expression data

We collected venom gene expression data for snakes from published literature that reported relative levels of toxin expression via transcriptome sequencing of cDNA libraries. From a list of 39 publications, we obtained data for a total of 52 different snake species from the three venomous

families (Colubridae, Elapidae and Viperidae). We included only species for which phylogenetic data were available, irrespective of transcriptome availability (see Table S1 in additional information (GitHub)), i.e. even if there were transcriptomic data available for a snake species, if the species was not present in our phylogeny, we excluded it. Our dataset included components that are found in at least 50% of the transcriptomes analysed here, this was done to focus on generally more widely abundant toxins (greater than 90% variation across 52 species) and because the sample sizes for the other components would be too low for accurate and phylogenetically unbiased inference, an approach similar to [27,50] (see Figs S1 and S2 in additional information (GitHub)). While changes in these toxins may well contribute significantly to the overall efficacy of venom, our goal was to trace the evolution of relatively ubiquitous components over time. As a result, our analyses are conducted on one component at a time, and the minor components do not greatly affect the percentage of the major components and thus do not affect the overall result [27, Supplementary Material]. Overall, 10 out of 27 toxins were retained (electronic supplementary material, figure S1). Viperid and elapid PLA2 are encoded by different loci and have evolved independently of one another [51]. Therefore, to make the interpretation of our data more intuitive for the reader, we classified elapid PLA2 (type I) as 'ePLA2' and viperid PLA2 (type II) as 'vPLA2'. The published time-calibrated phylogeny of squamates used in our study estimates the most recent common ancestor (root) of the three snake families at about 60 million years ago [37,52].

## (b) What are the evolutionary rate dynamics of venom toxins?

Key innovations are predicted to experience shifts in tempo and mode of evolution in response to changes in optima [9]. Along with this, we would expect transitions in evolutionary rates, with a key innovation experiencing subsequent reduction in evolutionary rates since the time of the first occurrence along a branch [4]. We used the Bayesian analysis of macroevolutionary mixtures (BAMM) [53], to determine shifts in rates of toxin expression evolution that took place at different points throughout the history of snake venom evolution, as well as changes in evolutionary rates over time.

For all the 10 toxin families, BAMM revealed several rate shifts along the phylogeny, indicating that evolutionary rates for toxins do not remain constant (phylorate plot, figure 1). Bayes factor estimates support the occurrence of at least one rate shift in all toxin families, indicating that toxin families have experienced changes in their evolutionary rates since becoming a part of the venom arsenal (see Fig. S4 in additional information (GitHub)). Changes in evolutionary rates since the common ancestor of venomous snakes denote different evolutionary trajectories of toxin families (figure 1). CRISP, SVMP and TFTx had a larger distribution of high evolutionary rates (warmer colours) in the ancestors of all venomous snake families and experienced subsequent slowdown in evolution rates (cooler colours) as modern species emerged. The remaining toxins start with slower rates of evolution, which eventually increased in extant species. The phylorate plots also provide configurations of rate change that explain the occurrence and distribution of

toxin families in venoms of modern snakes (figure 1). For example, SVMP shows a stark reduction in rates from the common ancestor of venomous snakes to elapid lineages, while it experiences increase in rates in viper lineages. TFTx shows the exact opposite trend, with an increase in elapids and reduction in vipers. BPP, vPLA2 and SVSP show rate trends consistent with their greater distribution in vipers.

Under the adaptive radiation hypothesis, ecomorphological rates should transition from rapid rates to slow, equilibrium rates as ecological niches get filled [54]. To identify these patterns, we estimated the rates of toxin expression evolution of each toxin family after the split of the three families. Our estimates of 'rate through time' revealed that toxin families show unique evolutionary rates and rate dynamics in each venomous snake family (figure 2).

PLA2s, SVMP, SVSP and TFTx, which make up the largest portion of the venom, have higher evolutionary rates than the other minor components. In colubrids, there was evidence of a delayed rate increase in CTL, BPP and ePLA2. KSPI and TFTx showed an increase in evolutionary rate, while SVMP showed a steady decline. Among colubrids, TFTx was the only toxin to experience an increase in evolutionary rate since divergence of the family.

In vipers, toxin families generally showed an increase in evolutionary rates, with a majority of them occurring at around the 20 Ma mark, just after the diversification of the major viperid lineages (figure 2), which is potentially consistent with venom evolution being linked to ecological opportunity. Two most abundant toxins in the vipers, i.e. SVSP and vPLA2, showed an increase in evolutionary rates since the divergence of viperid lineages, while SVMP showed a decrease.

In elapids, there were very few instances of increase in evolutionary rate. BPP, SVSP and SVMP showed rate increases likely due to their high expression in *Ophiophagus hannah*. The most widespread toxin family in elapid venom TFTx showed a decrease in rate. ePLA2 showed an almost steady rate at the origin of colubrids but experienced a jump around 35 million ago.

SVMP and TFTx show an interesting pattern where they seem to represent alternate venom types. SVMP has high rates and is dominant in vipers (and to an extent in colubrids), while TFTx has higher rates and is predominant in elapids. The alternate lineage of these toxins could be evidence of trade-offs, a pattern that we previously observed [27].

## (c) Which model of trait evolution best describes venom evolution?

We fitted a number of trait evolution models to our data to understand which evolutionary process best describes snake venom evolution. We tested BM, OU, EB and jump models (pulsed models) implemented in the pulsR package [20]. The Lévy process can be used to model jumps in trait evolution, which may be appropriate for traits like gene expression, which cannot be explained by simple stochastic models [22,55]. Furthermore, the REML estimation in pulsR can account for intraspecific variation in trait measurements, allowing a more robust model comparison (see Methods).

BM was used to model incremental phenotypic change based on stochastic changes in optima, while OU was used to model incremental evolution around a single optimum. The EB model aims to capture the slowdown in tempo over

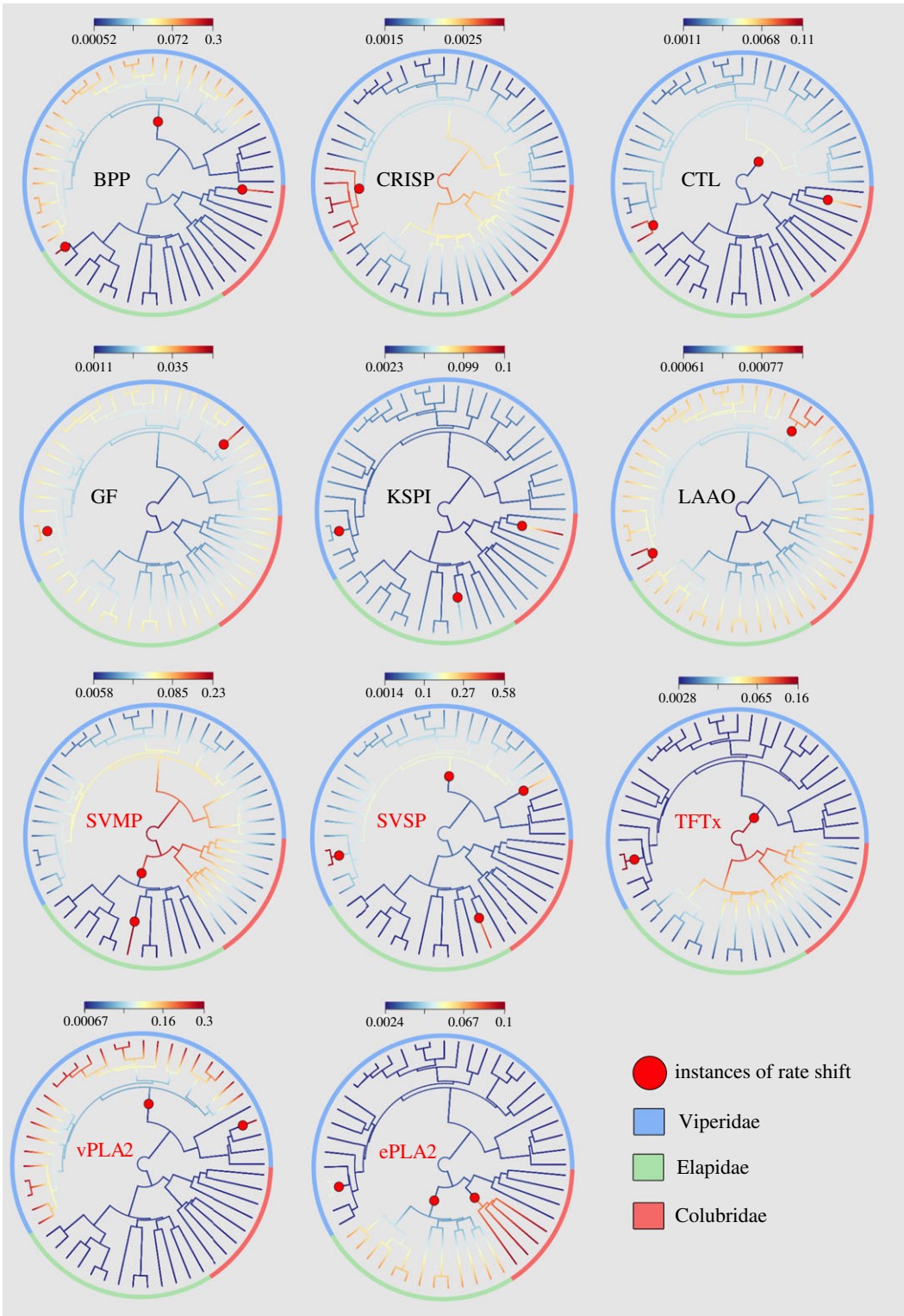

**Figure 1.** Snake venom phenotypes originated via multiple evolutionary rate shifts. BAMM phylorate plots show locations of the best rate shift configuration (red-filled circle) from among a large posterior distribution of shifts. Rate shift configurations are unique for each toxin family but all families experienced at least one rate shift, indicating a departure from the original evolutionary trajectory since the time of its first occurrence. The branches of the phylogeny are coloured based on distributions of evolutionary rates along the branch. Warmer colours denote a distribution of high evolutionary rates while cooler colours denote a distribution of low rates. With the exception of CRISP, TFTx and SVMP, all the other toxin families show slower rates near the root with a subsequent increase in modern snake lineages. (Online version in colour.)

time expected during adaptive radiation. Two variants of the jump process were modelled; jump normal (JN) and normal inverse Gaussian (NIG). The JN process represents infrequent jumps where stasis is followed by large-scale shifts in adaptive zones, while NIG represents more frequent jump processes, which captures the dynamics of constant phenotypic change that occurs by shifts within an adaptive zone [20]. Both these models represent a process of rapid-pulsed evolution. Jump models have the highest weighted AIC scores and are a better fit to snake venom gene expression data as compared with conventional incremental models of evolution (see Table S1 in additional information (GitHub)). The best model was one whose Akaike information criterion (AIC) weight was at least twice as high as other competing models. The

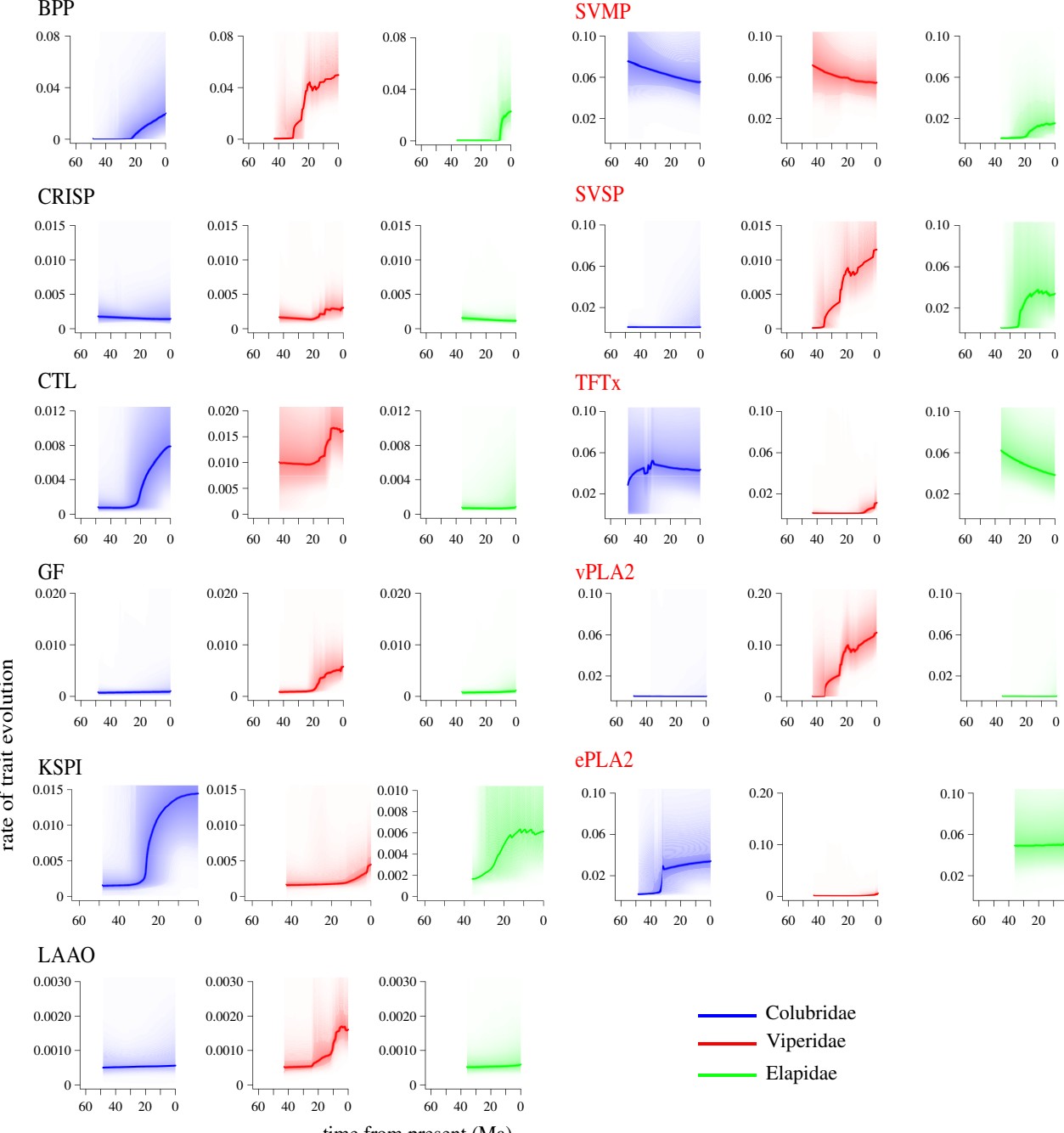

**Figure 2.** Family-specific trends in evolutionary rates of toxin expression can help explain variation in venom composition and toxin abundance observed between snake families. Blue, red and green (or leftmost, middle and right) represent evolutionary rates for Colubridae, Viperidae and Elapidae, respectively. The gradients represent confidence intervals for each of the estimated rates. The common trend in evolutionary rates between families is that they change through the course of venom evolution and different families have varying rates for individual toxins. (Online version in colour.)

jump models always have better fit than the incremental models; however, individual jump models (JN, NIG or any of their variants) do not differ much in their relative fit. To account for this, the Lévy process with the highest AIC weight was used to represent the entire class of jump processes.

Fitting our data to different models of phenotypic evolution showed that rapid-pulsed evolution explains the evolution of toxin gene expression better than incremental BM and OU process, or explosive EB process.

## 3. Discussion

One of the most intuitively appealing theories of how phenotypes influence long-term evolution of organisms is that of key innovations. Traits that provide ecological opportunity by allowing exploration of new potential niches, leading to adaptive radiation, seemed to be the perfect explanation of how trait evolution influences species evolution. But as mentioned before, this is far from the case. Many studies have shown that phenotypic disparity is only one of the many axes by which rapid radiation can take place [11]. Therefore, rather than looking at how specific traits correlate with species diversification rates, shifting focus towards identifying shifts in ecological and macroevolutionary space is a better representation of what key innovations achieve [9]. Our study found that gene expression in snake venom has experienced several shifts in evolutionary rates, and that rapid-pulsed evolution better explains snake venom

evolution (table 1). Both these results showcase the highly dynamic and rigorous process of venom evolution and remind us of its strong impact on both the ecology and evolution of venomous snakes.

## (a) Shifts in adaptive optima and rapid-pulsed evolution

The idea that snake venom evolution would be characterized by constant shifts in the evolutionary rates of toxin expression is not entirely unexpected. The high variability in the snake venom phenotype is likely due to the presence of various optima. A previous study showed that distribution of toxin families on the macroevolutionary scale can be explained by the presence of convergent phylogenetic optima [27]. Furthermore, the effect of various environmental factors like temperature and longitudinal climatic gradient influences venom variation, hinting at the occurrence of optima that maintain disparate, locally adaptive venom complexes [28]. Therefore, the shifts in phenotypic macroevolutionary rates are likely due to shifts between these optima. A combination of changes in prey diversity and changes in environmental conditions would lead to changes in adaptive optima, which would require snakes to constantly shift expression of different toxin families to chase the optima [28,56]. The extent of this combinatorial action towards diversification of traits in general is not known and might actually be restricted only to venom systems.

Rapid-pulsed evolution, on the other hand, is common in a variety of ecological, palaeontological and comparative data [20,57–59]. One of the proposed explanations for this kind of evolution is Wright's shifting balance theory [60]. The theory states that stochastic forces like genetic drift have a non-trivial effect on adaptation, and that populations occupying local adaptive peaks would compete with each other till the single fittest peak spreads to the entire species [60,61]. Considering the importance of population variation, genetic drift and adaptive peaks in snake venom divergence, the shifting balance theory could well be one of the explanations of how venom evolves [27,31,62].

Beyond the complicated dynamics of shifts in adaptive optima and punctuated evolution, looking at our results under a common perspective of toxin abundance, toxin age and evolutionary rate dynamics bring about an interesting evolutionary pattern. Abundant toxin families along with ePLA2 and SVSP showed a higher net rate of evolution of gene expression (figure 2), a trend also observed in sequence data [31,63]. This suggests that the most abundant (and often toxicologically dominant) toxins have the strongest links to the ecology and evolution of snakes and their venom systems—they are also probably more exposed to selection as a result.

Rate dynamics and age also tend to be related, with older toxins experiencing higher rates in the past followed by reduction in modern lineages (figure 1). It should be noted that the probable origin of most of the toxin families in our study pre-dates the root of our tree (electronic supplementary material, figure S1) [38]. Typically, if a trait is responsible for lineage diversification, its origin should be at a point within the tree around the time of major branching events. However, we can only examine what happens to venom evolution after the most recent common ancestor of extant snakes. Some of the oldest toxins to be included in the venom: SVMP, TFTx

**Table 1.** Rapid-pulsed evolution modelled as a Lévy process explain toxin expression evolution in snake venom better than conventional BM, OU and EB models. Model fits (weighted AIC) for BM, OU, EB and pulsed model of phenotypic evolution computed in pulsR [13]. Italic type indicates best fit. We use the AIC weight to determine which model best suits our data. The values in our table represent AIC weights for each of the nine models we tested (BM, OU, EB and six pulsed models). In all cases, the pulsed models were favoured as compared with the non-pulsed models. However, each pulsed model had very similar weights, which make it difficult to determine which pulsed model is better. For that reason, we club them together and report the highest AIC weight.

| toxin family | BM | OU | EB | pulsed |
|---|---|---|---|---|
| BPP | 0.039 | 0.014 | 0.051 | *0.377* |
| CRISP | 0.031 | 0.011 | 0.011 | *0.455* |
| CTL | 0.002 | 0 | 0 | *0.979* |
| GF | 0.171 | 0.062 | 0.062 | *0.465* |
| KSPI | 0 | 0 | 0 | *0.711* |
| LAAO | 0.123 | 0.021 | 0.122 | *0.335* |
| SVMP | 0.002 | 0 | 0 | *0.678* |
| SVSP | 0.127 | 0 | 0.046 | *0.349* |
| TFTx | 0 | 0 | 0 | *0.976* |
| vPLA2 | 0.089 | 0.033 | 0.032 | *0.363* |
| ePLA2 | 0 | 0 | 0 | *0.900* |

and CRISP [39], all showed a larger distribution of high evolutionary rates near the root than the tips (figure 2). This could be because the major toxin families were likely present in the ancestral venom and experienced a uniform reduction in evolutionary rates as lineages diversified. These toxins that pre-date the root likely allowed ancestral snake lineages to realize their ecological potential, which led to niche specification, which in turn led to a slowdown in evolutionary rates in their descendants. While it might be tempting to declare the above results evidence for trait-dependent diversification, one has to carefully look at all the possible lines of evidence, or rather lack thereof.

## (b) Could venom be responsible for adaptive radiation in venomous snakes?

The increase in abundance of different toxin families in snake venom likely provided a diverse range of phenotypic effects. While these diverse phenotypes are potential key innovations and can contribute to ecological opportunity by opening up previously unexplored feeding niches, they might not necessarily lead to adaptive radiation or show patterns of trait-dependent diversification [10,64]. For example, the Cocos finch (*Pinaroloxias inornata*), the only geospizine finch found outside the Galapagos, has colonized various feeding niches on Cocos Island, but has not speciated into different lineages [65]. This has been attributed to the fact that feeding differences alone did not lead to morphological or behavioural changes, and thus populations that have different feeding habits can still interbreed [66]. Trophic morphology (morphological characters related to food intake) in Lake Tanganyika cichlids provide only part of the impetus needed for rapid speciation, as body shape and microhabitat

traits are undergoing higher degrees of specialization to impart differences between species [67,68]. Even in the poster species for adaptive radiation, Darwin's finches, purifying selection to maintain optimal bill morphology influenced other behavioural, ecological and population dynamics, which prevented homogenization of breeding populations, aiding in speciation [69]. Therefore, when traits influencing feeding niche specification can lead to broader morphological, behavioural and ecological changes, speciation might occur.

Self-contained modular traits that evolve independently of each other can actually reduce the potential of a species to attain large-scale diverse forms. For example, species with highly modular traits can individually evolve different aspects of those traits, without having a large influence on the overall biology of the animal [10,70]. The venom system comprises venom toxins, venom glands, fangs and muscle architecture responsible for delivering the venom into the prey. Numerous examples exist of toxin recruitments coinciding with the development of various morphological features like high-pressure venom delivery and certain hunting strategies like ambush feeding [39]. However, any modifications to enhance prey procurement would be restricted to the venom system and unlikely to affect changes in other parts of the animal [25]. In Darwin's finches, modification of bill morphology influences mating behaviours, where females do not choose males whose bill morphology starkly differs from theirs [69,71]. It is not known if snakes exhibit mating preference based on venom composition or related adaptations. For example, would a female prefer a male with more similar or dissimilar venom composition for mating? Venom might lead to indirect ecological consequences in terms of foraging style, habitat choice and temporal differences in activity. But speciation requires a level of reproductive isolation; how this is achieved either directly or indirectly by changes to the venom is not obvious.

# 4. Conclusion

Studies of adaptive radiation and character evolution are complex and often come with several caveats. Nearly all studies of adaptation focus on traits and processes in extant species, and this is a major disadvantage since there is no way of representing extinct taxa and thus no way of determining whether a clade with specific innovations was more species-rich in the past [72,73]. While most studies provide a microevolutionary perspective, extrapolating from processes that operate in the present day to what happened early in a clade's history is difficult; because conditions were different in the past, different processes may have been at work or may have produced different outcomes [73]. Perhaps in the past there were venomous snakes with venom compositions specific to the past environment. In response to any changes in this environment, snakes could have evolved venom compositions starkly different from the ones we see today. There might also have been venomous snake lineages in the past that became extinct, leaving a whole history of venom composition unexplored.

The selective and adaptive advantages of snake venom are in no doubt, and based on our results, venom in snakes can be rightly classified as a key innovation. Snakes usually need to produce large amounts of venom, and determining if venom is costlier compared with other offensive (or defensive strategies) is difficult, as it requires prey-handling experiments, taxon-specific toxicity testing, etc., which are both complicated, and difficult to implement [41]. However, considering the several ways snakes can modulate venom output (e.g. venom metering, secretions with reduced protein content etc.), venom might actually be an effective way of procuring energy-rich meals (by subduing large prey), making it a particularly cost-effective innovation [41,74]. Despite this, we believe venom is not the sole reason for the radiation of venomous snakes.

Key innovations are not the only sources of ecological opportunity. The effect of new habitat, antagonistic extinction and key innovations act in concert to promote ecological release which leads to adaptive radiation [3]. Therefore, key innovation plays only one part in the triumvirate of ecological opportunity; if the relative impacts of new habitat, antagonistic extinction and early-stage allopatry are sufficiently strong, release from natural selection and subsequent adaptive radiation might still take place [64,75,76]. Adaptive radiation is also subject to certain initial conditions. Some clades tend to radiate more than others, suggesting that evolvability and the propensity to speciate are vital for adaptive radiation to take place [10]. Looking at the evolution of snake venom in terms of its impacts on speciation would provide greater insight into the role of snake venom in adaptive radiation of venomous snakes. As venom transcriptomes of more snakes become available, revisiting our workflow would tell us to what extent our results represent a general trend in evolution of gene expression in snake venom.

# 5. Material and methods

## (a) Data collection and phylogenetic tree

We used a dataset comprising 10 toxins, which account for greater than 90% of the total venom composition across 52 snake species. The data were collected from a list of 39 publications (see Table S1 in additional information (GitHub)). We included only species for which phylogenetic data were available, irrespective of transcriptome availability (see Table S1 in additional information (GitHub)), i.e. even if there were transcriptomic data available for a snake species, if the species was not present in our phylogeny, we excluded it. We scaled gene expression levels by the average within-species variance, allowing us to standardize the measurements and carry out comparisons across species. This scaled dataset was used in all subsequent analysis. We used a previously described time-calibrated phylogeny of squamates based on two large datasets comprising 44 nuclear genes for 161 squamates, and a dataset of 12 genes from 4161 squamate species; both these datasets represented families and subfamilies [27,37,52,77]. While we manage to sample the three main families of venomous snakes, how under-sampling some species affects our analysis has been discussed in the 'Analytical considerations' section in the additional information (GitHub).

## (b) Evolutionary rate dynamics

We used BAMM [53] to estimate evolutionary rate dynamics for each toxin. We ran BAMM on normalized toxin values for each toxin family. We modified the BAMM control file to carry out analysis for phenotypic evolution. *Modeltype* was set to 'trait' and was run for $10^9$ generations with MCMC write frequency of $10^5$. Priors were obtained using the *setBAMMpriors* function in BAMMTools [78]. For our analysis, we used a conservative prior with *expectedNumberOfShifts* = 1; this model assumes zero rate shifts will have a higher prior probability. Using the

Bayesfactor calculations implemented in BAMMTools, we can determine if the rate shifts we obtain are significantly different from a model with zero rate shifts (or the lowest possible rate configuration where zero rate shifts cannot be computed). The convergence of MCMC chains was determined by visual inspection, by plotting *effective sample size* of log-likelihood and number of shifts in each sample, both of which well exceeded the recommended value of 200 (see Fig. S3 in additional information (GitHub)). We used the *credibleShiftSet* function to identify 95% credible set of distinct shift configurations (see additional information (GitHub)). The rate configuration reported in figure 1 was obtained using the *getBestShiftConfiguration* command in BAMMTools. Explanations behind two rate configurations potentially misrepresented in the phylorate plots are provided in the 'Analytical considerations' section in the additional information (GitHub).

## (c) Trait evolution models

We used the pulsR package to fit classes of evolutionary models [20]. Standard variants of incremental evolution BM, OU and EB were modelled as a BM process with branch lengths rescaled as a function of the model parameters [79]. Pulsed evolution, on the other hand, was modelled as a Lévy process. The Lévy process is a stochastic process characterized by three components: (i) a constant directional drift $\mu$, (ii) a Brownian motion with rate $\sigma$, and (iii) a jump measure $v(dx)$. The Lévy process is represented mathematically using the Lévy–Khinchine representation, where one can compute the variance of trait change along a branch of length $t$. We model stasis followed by rapid adaptation using a compound Poisson process. This is the JN process, which assumes jump sizes are drawn from a normal distribution. The other pulsed evolution model is NIG, which uses an infinitely active Lévy process to model constant rapid adaptation. Since a trait measurement is usually a statistic of a population (trait mean), its value cannot be exactly known. For this reason, the REML estimation assumes that observed traits are drawn from a normal distribution around their true values. This is modelled as a 'tip noise' parameter ($\sigma_{tip}$). This parameter is estimated as a combination of both sampling error due to intraspecific variation as well as measurement error. The parameter $\sigma_{tip}$ can be used as a proxy for $\sigma_{intra}$, and Landis & Schraiber [20] have shown that $\sigma_{tip}$ predicts $\sigma_{intra}$ moderately well. Weighted AIC was used as a measure of model fit. We decided to favour a particular model only if its Akaike weight is at least twice as high as its competing model, similar to [20]. While this arbitrary criterion indeed lacks elegance, it makes model comparison easier and removes ambiguity, especially considering that we are comparing many models. As the AIC between various jump models does not differ greatly, we clubbed them (JN + NIG + BMJN + BMNIG + EBJN + EBNIG) together, and the highest AIC weight was used to represent the entire class of jump processes.

Data accessibility. Additional information, comprising figures, tables, datasets, original plots, code and a section about analytical caveats, can be found at https://agneeshbarua.github.io/venom-phenotype-evolution/. AIC and code for Lévy models can be found at: https://agneeshbarua.github.io/LevyModels/.

Authors' contributions. A.B. and A.S.M. conceptualized the study. A.B. collected the data and carried out the analysis. Both A.B. and A.S.M. wrote the paper.

Competing interests. We declare we have no competing interests.

Funding. We received no funding for this study.

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
