## [Reviewer comments · Proceedings of the Royal Society B: Biological Sciences]

Review History

RSPB-2020-0122.R0 (Original submission)

Review form: Reviewer 1 (Kevin Arbuckle)

Recommendation

Major revision is needed (please make suggestions in comments)

Scientific importance: Is the manuscript an original and important contribution to its field?

Good

General interest: Is the paper of sufficient general interest?

Good

Quality of the paper: Is the overall quality of the paper suitable?

Good

Is the length of the paper justified?

Yes

Should the paper be seen by a specialist statistical reviewer?

No

Do you have any concerns about statistical analyses in this paper? If so, please specify them explicitly in your report.

No

It is a condition of publication that authors make their supporting data, code and materials available - either as supplementary material or hosted in an external repository. Please rate, if applicable, the supporting data on the following criteria.

Is it accessible?

Yes

Is it clear?

Yes

Is it adequate?

Yes

Do you have any ethical concerns with this paper?

No

Comments to the Author

Firstly, I should disclose that I have already reviewed this manuscript elsewhere on a prior submission (to my knowledge this is no longer being considered there so there is no conflict), and I am glad to see that substantial changes have been made which, on the whole I think have improved the manuscript. Nevertheless, I still have outstanding concerns/comments here, partly issues that remain from my earlier review and partly new comments introduced in the current version. I think this manuscript is well placed to make an impact in the field if these comments are addressed, and so I list them below (in roughly chronological order) in the hope that they are useful to the authors.

Ln 41-42 - I think you need to clarify your comment that there have been 'no quantitative analyses' testing whether venom is a key innovation. There have been tests of the hypothesis that venom is associated with higher diversification rates, which is often regarded as a major component of key innovations and, more recently, has been regarded as a non-essential but (nevertheless) a common outcome of key innovations (and by many definitions is the 'radiation' part of adaptive radiation).

Ln 51 - I'm not aware of any cases where someone has proposed that venom acts as a 'magic trait', merely as a key innovation.

Ln 108-111 and 119-120 - I'm not convinced by the relevance of your assertion that "highly tissue specific genes would also likely reduce significant pleiotropic constraints and cross-phenotype associations helping to discern the unique trajectories of individual genes". While tissue-specific expression would certainly reduce such interactions across tissues, many venom toxins are widely suspected or known to have lots of pleiotropic functional effects and synergistic or antagonistic interactions with other venom components. In other words, as a highly specialised and multidimensional (complexity of different interacting toxins) system I don't think that venom molecular evolution can be expected to typically have low rates of pleiotropy or molecular interactions.

Ln 128-129 - Depending on your definition of key innovation you could perhaps argue that phenotypic differentiation driven by ecology has not been studied well in snakes (although you could easily consider some of the work on diet and venom evolution to fall into this bracket), however, when talking about "the extensive radiation of snakes", which strongly implied lineage diversification, it is untrue to claim that "no studies have tested this hypothesis". In fact, I find it surprising that none of the work linking venom evolution to lineage diversification rates appears

in this paper given it's clear relevance at various points. In the interest of full disclosure, some of this work is my own (e.g. Harris and Arbuckle, 2016, *Toxins* 8:193) but other work is not (e.g. Liu et al., 2018, *Mol Phylogenet Evol* 125:138-146).

Ln 135 - optima for what?

Ln 145-147 - How did you deal with cases of multiple studies of the transcriptomes for a single species?

Ln 159 - Fig 1 seems to be missing. I am unclear from the wording here whether you are referring to Fig 1 within the Supplementary Material or both Fig 1 and Supplementary Material, but in either case there is no Fig 1 in your manuscript (numbering starts at Fig 2).

Ln 163-165 - What is the relevance to your study of age of the clade in your study? It seems an odd detail to add with no explanation of its relevance.

Ln 180 - "toxins families" should be 'toxin families'

Ln 183-192 - There are a few points of interpretation of your data that I'm not sure you have considered (at most having a couple of partially related comments later in the manuscript for some of these points but not fully incorporated or explored). Of the toxins with the best evidence of 'early burst' patterns (which are potentially indicative of adaptive radiation) two important toxins stand out - SVMs and TFTs. Specifically, these are often the most abundant toxins, are often toxicologically dominant ones, and the 'early burst' patterns are largely present in vipers (for SVMs, colubrids also show this pattern) and elapids (for TFTs, also stated on Ln 211), which also corresponds to the main toxin classes in each of these lineages. The alternative lineages for these toxin patterns may be evidence for trade-offs (as shown previously by the authors - in practice if not necessarily these toxin classes appear as 'alternative' venom composition types). Importantly, the time-scale of toxin origins in relation to your clade under study is important for interpreting patterns of evolutionary rate dynamics. In your results, some toxins start off with very low rates and show patterns consistent with early burst style dynamics (e.g. C-type Lectins and SPs) or maintain low rates across most of the tree (e.g. KSPIs). As these are predicted to pre-date the root of your tree they may have already experienced an early burst before slowing prior to the earliest point of your study. Consistent with this idea is that the end of the early burst style patterns of SVM and 3FTs have estimated rates comparable to the root estimates of other toxins. For some of the patterns you are looking for it is important to have the origin of the trait within the phylogeny and this important caveat is neither stated nor discussed (the closest is on Ln 296-298 but this failed to acknowledge that almost all the toxin families here pre-date your clade, and most of them originated on one of two branches).

Ln 195-197 - How are you defining "major lineages" in the context of 'expectations that rates of trait evolution should be high immediately after they diverge'? Why is the split of the three venomous families an important evolutionary point for your study (note also that you should check if 'Colubridae' is actually the family you mean as this 'traditional wastebucket taxon' has been split into many families)? What's special about the divergence of these particular clades compared to any other clades on the tree?

Ln 201 - Colubrids shouldn't be capitalised

Ln 206-207 - This time period is approximately coincident with the colonisation of the New World by vipers, which is potentially consistent with venom evolution being linked to ecological opportunity.

Ln 207-208 - You claim that "SVM...showed an increase in evolutionary rates since the divergence of vipers", but Fig 3 seems to contradict this.

Ln 210 - "Ohiophagus hannah" should be 'Ophiophagus hannah'

Ln 219 - There should be an 'and', not a comma between "Brownian motion (BM)" and "Ornstein-Uhlenbeck (OU)".

Ln 220 - Why isn't Early burst a 'conventional' model? It's been a fairly standard part of the set of phenotypic evolution models for a pretty long time now.

Ln 292-294 - This again suggests that the most abundant (and often toxicologically dominant) toxins are likely the most important links to the ecology and evolution of the animals and their venom systems - they are probably more exposed to selection as a result.

Ln 298-299 - You didn't estimate ancestral character states (or at least don't present them in the manuscript), you estimate rate dynamics.

Ln 306-307 - Why not test for trait-dependent diversification rather than simply say 'it is tempting to speculate on this but there is a lack of evidence' (and, as above, you might want to discuss what evidence does exist for venom/toxin-associated diversification). You again say something similar on Ln 314-315 in that venom 'might not necessarily...show patterns of trait-dependent diversification', well, not necessarily but you could test for it.

Ln 312-313 - phenotypes don't lead to key innovations, phenotypes are the key innovations (the evolutionary consequences of which are what makes the difference to terminology)

Ln 329 - What do you mean by 'large-scale' diverse forms?

Ln 338 - It is very unlikely that snakes show mate choice based on venom composition, but that very direct mechanism is not necessary for venom phenotypes to "cause widespread character changes that are needed to establish reproductive isolation that leads to speciation". Indirect consequences would still ultimately be due to venom variation, for instance ecological separation (complete or partial) relating to diet specialism, foraging style, related habitat choice (for particular prey), differences in activity (better defended species may be less constrained by predation, be more active, and hence achieve a higher frequency of mating opportunities). As you allude to, little direct evidence for any of these types of mechanisms exist, but the important part is that neither direct venom-based mate choice nor widespread character changes (it could be one or two simple but important changes) are necessary.

Ln 346 - I think you mean 'adaptation' or 'adaptive radiation' instead of just "adaptive".

Ln 347-348 - Although you are correct that there is no way of "determining" (with absolute certainty) whether a clade with particular traits was more diverse in the past, there are plenty of ways to estimate this, so it's not quite so much of a lost cause as you suggest.

Ln 351-352 - Can you suggest plausible examples of such different conditions and processes that are relevant to venom evolution? Unless we have evidence of different processes we typically assume some degree of uniformitarianism (without which any historical science is impossible).

Ln 354 - Should "analysis" be 'analyses'?

Ln 355-356 - Just because you are necessarily basing your interpretations on the data you collected doesn't mean that your interpretations and findings are specific to your dataset only...at least I hope not as this would undermine the biological insights your study could give (or indeed any study ever published).

Ln 360-362 - I would have thought that the locations of rate shifts are far more relevant to your hypotheses than the number of rate shifts? Would your hypotheses be better supported if you

found 6 vs 3 vs 1 shifts at random points in the phylogeny or particular rate shifts that occur at locations consistent with related evolutionary events you think are important?

Ln 371 - "misrepresented" should be 'misrepresented'

Ln 374 - you have doubled up the words "by the"

Ln 384-388 - I don't follow why a procedure applied to all species would only affect this one particular case.

Ln 403-405 - I agree that considering "the evolution of snake venom in terms of its impacts on speciation" would provide insight into venom as an influence in adaptive radiation, but as mentioned earlier, this has been tested in some other venomous groups (and also including, in combination with other tetrapods, in snakes). The existing studies focus on the presence of venom, rather than specific attributes, so there is more to be done here, but I think the lack of any discussion of this literature is a bit of an oversight (given you refer to the basic questions it aims to address at multiple points in your manuscript).

Ln 421-424 - How did you combine these two trees, and why did you choose to use two different phylogenies?

Ln 428-429 - something has gone a bit wrong with superscripts here.

Ln 430 - "conservatiove" should be 'conservative'

Ln 433 - By "significant from" do you mean 'significantly different from'?

Ln 434 - Convergence doesn't need to be capitalised.

Ln 443 - What do you mean by an "explosive EB" model? How is this different from any other EB model?

Ln 449-450 - I'm not sure what the sentence beginning with "one of the models" is saying, the wording is very confusing.

Ln 459 - When you say "weighted AIC was used as a measure of model fit" I'm not sure what you mean - do you mean AICc (which isn't mentioned elsewhere) or Akaike weights (aka model probabilities, which looks more similar to what is presented in ESM Table 1, but in this case I don't understand why the values in that Table don't sum to 1 in each row)? Particularly if you mean Akaike weights, these aren't a "measure of model fit" as such, but merely the probabilities of each model being the best in your model set (they say nothing about how good your model set is - none of them could fit well but you'll still get some better than others).

Ln 460-461 - The absolute values of AIC are meaningless for interpretation, and hence their ratio is as well, so I assume it's just unclear what you mean here but a model with twice the AIC of another model says nothing about the relative evidence each provides.

I hope the authors aren't too discouraged by these comments, and (as I said earlier) the manuscript is certainly in better shape now than the earlier version I reviewed. I do believe the basic idea and analyses here have something important to offer the field, but my comments are intended to help improve the presentation and interpretation of these results. I hope the authors find them useful.

Best wishes,
Kevin Arbuckle

Review form: Reviewer 2 (John P. Hunter)

Recommendation

Accept with minor revision (please list in comments)

Scientific importance: Is the manuscript an original and important contribution to its field?

Excellent

General interest: Is the paper of sufficient general interest?

Excellent

Quality of the paper: Is the overall quality of the paper suitable?

Excellent

Is the length of the paper justified?

Yes

Should the paper be seen by a specialist statistical reviewer?

No

Do you have any concerns about statistical analyses in this paper? If so, please specify them explicitly in your report.

No

It is a condition of publication that authors make their supporting data, code and materials available - either as supplementary material or hosted in an external repository. Please rate, if applicable, the supporting data on the following criteria.

Is it accessible?

Yes

Is it clear?

Yes

Is it adequate?

Yes

Do you have any ethical concerns with this paper?

No

Comments to the Author

Review of RSPB-2020-0122

General Comments

This is an interesting and significant investigation of the macroevolutionary consequences of possessing venom in snakes. The study infers rates and patterns of evolution in toxin gene expression through transcriptome techniques. The data are mined from published work and assembled and analyzed here to test various evolutionary pattern and process models for goodness of fit. In the end, the authors demonstrate that pulsed models, in which rates of evolution vary through time as if in response to shifting optima, fit the data better than random and early burst models. The core of the study is bracketed between an Introduction and a Discussion and Conclusion interpreting venom as a key innovation within a framework of

adaptive radiation.

Overall, the paper is well written, genuinely compelling, and shows an understanding of many kinds of data from multiple subfields of evolutionary biology. There is something in this paper for everyone. The authors demonstrate an awareness of and acknowledge the limitations of their own data. They also are aware of the strengths of other kinds of data.

The overall rationale for the study, as presented here, is to test whether gene expression data conform to an early burst model of evolution better than phenotypic data. The authors present the early burst model as the model expected of an adaptive radiation resulting from a key innovation that facilitates crossing an adaptive threshold and entering a new adaptive zone, followed by slower rates with the filling and subdivision of ecological niches. This view fits in with a recently proposed, and somewhat restrictive stance on key innovations. In the end, the authors found poor fit of venom toxins to the early burst model and better fit to pulsed models. This result implies that venom toxins do not just enable entering a new adaptive zone. Rather venom toxin continues to evolve with rapid pulsed episodes and to play a major role in the adaptations of snakes for capturing prey, perhaps enabling the coexistence of species, which supports higher standing diversity. In this respect, snake venom is not an anomaly, but rather is within a class of key innovations that promote further evolutionary change. Such adaptations are evolutionarily versatile in the sense of being highly evolvable and able to adapt quickly to changing conditions. I feel that the authors could make a more streamlined case in their Introduction if they took a more open-ended view of key innovations from the onset and discussed the evolutionary models (early burst and pulsed) that are consistent with scenarios on how different kinds of key innovations might operate. Doing so changes the focus from whether venom is a key innovation, which ultimately depends on one's definition of the term, to how venom has evolved and influenced the evolutionary radiation of venomous snakes.

Specific Comments

Title

The title summarizes the main results of this study about the tempo and mode of toxin evolution in snakes, but it does not fully convey what the paper is really about, which is whether snake venom is a key innovation and how it has influenced the radiation of snakes. Perhaps the title ought to reflect the full scope of the paper.

Abstract

Line 40. "are essential for prey capture" should perhaps be "facilitate prey capture". The latter wording is more active. Also, venom is essential only if a snake is restricted to a certain predation strategy (strike, follow, and wait) that requires it. Other snakes (boas and pythons) can also capture prey without venom, but with a different strategy and adaptations.

Line 51. Change "influence" to "influences". Insert "in" after "process" and "that" after "way". Also, "magic traits" is an odd phrase that requires explanation or at least a reference. Therefore, it may be better used in the main text than here in the abstract where it can neither be explained nor a reference cited.

Introduction

Line 60. "ecological speciation". Some authors (e.g., Mike Rosenzweig) have used this term to mean a kind of sympatric speciation, whereas other do not and treat it as a kind of allopatric speciation. The authors should probably explain what they themselves mean by using the term.

Lines 67-70. "Rather, the role of key innovations should be restricted to providing entry into novel ecological niches or adaptive zones, and studies should aim to identify specific shifts in

tempo and mode of phenotypic evolution of the assumed key innovation (9,10).” If the authors wish to adopt this restricted definition of key innovation, or more precisely criteria for recognizing a key innovation, that is fine. However, if adopting this definition leads them to use the Early Burst model as the sine qua non of a key innovation, as they seem to do in the following paragraph, then they undermine their own argument. Ultimately, the authors find that the evolution of venom toxins in snakes is better explained by Pulsed models than by an Early Burst model. That does not surprise me because venom toxin seems like a evolutionarily versatile system as the authors describe it, a lot like the pharyngeal jaws of cichlid fishes in sense of facilitating further, rapid evolutionary change. Finding pulsed rate of evolution is what I would expect of a key innovation that promotes further evolutionary change. The next thing that I would want to know is what are the tradeoffs. Are there any costs to having venom? How about of certain combinations of toxins? I would advise the authors to frame their argument about venom as a key innovation in comparison to other innovations that are versatile and highly evolvable. Such adaptations tend to be those with low costs to the organisms, often through relaxed trade-offs among functions.

Line 85. “rarely observed in data (14).” The authors should clarify what kind of data. Comparative data in extant clades, I should think. In the fossil record, the early burst pattern is more commonplace but by no means universal.

Lines 85-99 “This lack of empirical support might be because studies are overlooking the components of traits that actually produce phenotypic change; gene expression variation” and continuing to the end of the paragraph. I did not find this particular argument to be a convincing explanation for why few examples of early burst adaptive radiations occur among extant species. I do understand the authors’ main point that rates of evolution of genes and phenotypes may differ, and I understand the authors’ need to motivate their study around a specific question and a hypothesis. However, in adaptive evolution genes are only going to evolve as rapidly as the phenotypes to which they correspond are sorted by natural selection. One way in which genes might evolve faster is in drift among selectively neutral variants. But probably would not the case here. Alternatively, and perhaps more of a problem, is in cases of highly developmentally canalized systems in which the same phenotype can develop from more than one, and perhaps multiple different gene expression patterns. This situation may be commonplace and could lead to higher rates of evolution in genes than in phenotype.

Lines 100-127. The problem of gene expression-phenotype mapping in complex traits versus in snake venom. The authors do make a good case here for the special status of venom because of its simplicity, in which different levels of gene expression result directly in different proportions among toxins in the venom “cocktail”. In other words, in snake venom one need not consider gene networks, morphogenetic pathways, and the like. The system sounds ideal. However, several thoughts occurred to me as I read this passage. What are the observed performance differences among toxin cocktails? What specific associations exist between toxin mixtures and either prey or environments? The idea that toxins do not interact with other is presented rather definitively and with citation of just one supporting reference. How was this inference made? Although the authors present toxins within venom as isolated adaptations without interactions, venom requires a delivery system (glands and fangs) as they later mention briefly, a specific hunting strategy (behavior and supporting morphology, including sensations), and either immunity from the effects of one’s own toxins, some barrier sequestering toxins, or at least storage of toxins in an inactive state until deployed. How is all this potential complexity managed? One could imagine far-reaching consequences of having venom or of variation among venom cocktails on the phenotypes of snakes. Is venom really such an isolated system?

Lines 133-135. “If snake venoms are key innovations we would expect to see shifts in their evolutionary rates in response to changes in optima.” How are optima identified? Ideally, this would be by reference to evidence external to the snakes themselves, such as the timing of environmental changes from the paleoclimatic record, biogeographic dispersal events into new regions, or time of contact with novel prey. There is a certain circularity to inferring the existence

of optima from the evolutionary rate dynamics of snake venom alone. Is there external evidence too?

Results

Lines 232-235. "Jump models have highest weighted AIC scores and are a better fit to snake venom gene expression data as compared to conventional incremental models of evolution (Table 1; electronic supplementary material)." Are the jump models grouped together in Table 1 under the heading "Pulsed"? If yes, then please state so. In general, yes, the "Pulsed" models all fit better than the random (BM and OU) and early burst (EB) models. Do the pulsed models fit significantly better? Also, can the "pulsed" models distinguish oscillation between stasis and random walk, on the one hand, from sporadic pulses of directional change, on the other hand? In order to relate these models to those traditionally used to study evolutionary rates in paleontology (stasis, random walk, and directional evolution), it would be beneficial if the authors would explain the correspondences between these models and traditional ones.

Lines 273-281. Optima, prey diversity, and environment. So far, the existence of optima has only been inferred in the authors' data from the good fit of the data to pulsed (Lévy process) models that assume optima. The authors could make a stronger argument if they could be more specific about what the optima are and how venom has evolved specifically to allow snakes to adapt to these shifting optima. If specific associations are lacking, then at the very least they could explain how prey diversity and environment could in theory co-define optima as the authors claim. For example, what toxin combinations are better suited to a diverse range of prey, and which are better suited to one or a few prey types? Do venomous snakes conform to the specialist/generalist dichotomy? Are there tradeoffs between toxin efficacy on one kind of prey and the number of prey types on which it can be used? How does environment factor in? Through general levels of metabolic and behavioral activity expected of an ectotherm? Or in more specific ways through prey availability, overall or seasonally? Right now, these optima are just a little too vague to be totally satisfying.

Lines 346-349. "Nearly all studies of adaptive [change to "adaptation"] focus on traits and processes in extant species, [and] this is a major disadvantage since there is no way of representing extinct taxa and thus no way of determining whether a clade with specific innovations was more species[-]rich in the past (77,78)." Well said, and very refreshing to see this statement here. Please note minor edits in brackets.

Decision letter (RSPB-2020-0122.R0)

21-Feb-2020

Dear Mr Barua:

I am writing to inform you that your manuscript RSPB-2020-0122 entitled "Toxin expression in snake venom evolves rapidly with constant shifts in evolutionary rates" has, in its current form, been rejected for publication in Proceedings B.

This action has been taken on the advice of referees, who have recommended that substantial revisions are necessary. With this in mind we would be happy to consider a resubmission, provided the comments of the referees are fully addressed. However please note that this is not a provisional acceptance.

The resubmission will be treated as a new manuscript. However, we will approach the same

reviewers if they are available and it is deemed appropriate to do so by the Editor. Please note that resubmissions must be submitted within six months of the date of this email. In exceptional circumstances, extensions may be possible if agreed with the Editorial Office. Manuscripts submitted after this date will be automatically rejected.

Sincerely,
Dr Sasha Dall
mailto: proceedingsb@royalsociety.org

Associate Editor
Board Member: 1
Comments to Author:
Dear Authors,

First, apologies for the long time it took to get the reviews in: the blame should not be put on the reviewers in this case, but on me: both of the reviewers returned their reports well in time.

As you can see from the referee reports, both reviewers think highly of your work, and believe that it could make an important contribution to the field. At the same time, both of them raise number of concerns, which I hope you would able to deal in revision. However, given the quite substantial requests for clarification and re-writing, it is hard for me to provide the option for revisions, but I would be glad to handle a new review process of a revised version.

Even if critical, I hope that you find the reviewers' comment useful - their general tone is very constructive: both reviewer's recognise that value of your work.

Best wishes,
Juha Merilä

Reviewer(s)' Comments to Author:

Referee: 1

Comments to the Author(s)

Firstly, I should disclose that I have already reviewed this manuscript elsewhere on a prior submission (to my knowledge this is no longer being considered there so there is no conflict), and I am glad to see that substantial changes have been made which, on the whole I think have improved the manuscript. Nevertheless, I still have outstanding concerns/comments here, partly issues that remain from my earlier review and partly new comments introduced in the current

version. I think this manuscript is well placed to make an impact in the field if these comments are addressed, and so I list them below (in roughly chronological order) in the hope that they are useful to the authors.

Ln 41-42 - I think you need to clarify your comment that there have been 'no quantitative analyses' testing whether venom is a key innovation. There have been tests of the hypothesis that venom is associated with higher diversification rates, which is often regarded as a major component of key innovations and, more recently, has been regarded as a non-essential but (nevertheless) a common outcome of key innovations (and by many definitions is the 'radiation' part of adaptive radiation).

Ln 51 - I'm not aware of any cases where someone has proposed that venom acts as a 'magic trait', merely as a key innovation.

Ln 108-111 and 119-120 - I'm not convinced by the relevance of your assertion that "highly tissue specific genes would also likely reduce significant pleiotropic constraints and cross-phenotype associations helping to discern the unique trajectories of individual genes". While tissue-specific expression would certainly reduce such interactions across tissues, many venom toxins are widely suspected or known to have lots of pleiotropic functional effects and synergistic or antagonistic interactions with other venom components. In other words, as a highly specialised and multidimensional (complexity of different interacting toxins) system I don't think that venom molecular evolution can be expected to typically have low rates of pleiotropy or molecular interactions.

Ln 128-129 - Depending on your definition of key innovation you could perhaps argue that phenotypic differentiation driven by ecology has not been studied well in snakes (although you could easily consider some of the work on diet and venom evolution to fall into this bracket), however, when talking about "the extensive radiation of snakes", which strongly implied lineage diversification, it is untrue to claim that "no studies have tested this hypothesis". In fact, I find it surprising that none of the work linking venom evolution to lineage diversification rates appears in this paper given it's clear relevance at various points. In the interest of full disclosure, some of this work is my own (e.g. Harris and Arbuckle, 2016, *Toxins* 8:193) but other work is not (e.g. Liu et al., 2018, *Mol Phylogenet Evol* 125:138-146).

Ln 135 - optima for what?

Ln 145-147 - How did you deal with cases of multiple studies of the transcriptomes for a single species?

Ln 159 - Fig 1 seems to be missing. I am unclear from the wording here whether you are referring to Fig 1 within the Supplementary Material or both Fig 1 and Supplementary Material, but in either case there is no Fig 1 in your manuscript (numbering starts at Fig 2).

Ln 163-165 - What is the relevance to your study of age of the clade in your study? It seems an odd detail to add with no explanation of its relevance.

Ln 180 - "toxins families" should be 'toxin families'

Ln 183-192 - There are a few points of interpretation of your data that I'm not sure you have considered (at most having a couple of partially related comments later in the manuscript for some of these points but not fully incorporated or explored). Of the toxins with the best evidence of 'early burst' patterns (which are potentially indicative of adaptive radiation) two important toxins stand out - SVMs and TFTs. Specifically, these are often the most abundant toxins, are often toxicologically dominant ones, and the 'early burst' patterns are largely present in vipers (for SVMs, colubrids also show this pattern) and elapids (for TFTs, also stated on Ln 211), which also corresponds to the main toxin classes in each of these lineages. The alternative

lineages for these toxin patterns may be evidence for trade-offs (as shown previously by the authors - in practice if not necessarily these toxin classes appear as 'alternative' venom composition types). Importantly, the time-scale of toxin origins in relation to your clade under study is important for interpreting patterns of evolutionary rate dynamics. In your results, some toxins start off with very low rates and show patterns consistent with early burst style dynamics (e.g. C-type Lectins and SPs) or maintain low rates across most of the tree (e.g. KSPIs). As these are predicted to pre-date the root of your tree they may have already experienced an early burst before slowing prior to the earliest point of your study. Consistent with this idea is that the end of the early burst style patterns of SVMP and 3FTxs have estimated rates comparable to the root estimates of other toxins. For some of the patterns you are looking for it is important to have the origin of the trait within the phylogeny and this important caveat is neither stated nor discussed (the closest is on Ln 296-298 but this failed to acknowledge that almost all the toxin families here pre-date you clade, and most of them originated on one of two branches).

Ln 195-197 - How are you defining "major lineages" in the context of 'expectations that rates of trait evolution should be high immediately after they diverge'? Why is the split of the three venomous families an important evolutionary point for your study (note also that you should check if 'Colubridae' is actually the family you mean as this 'traditional wastebucket taxon' has been split into many families)? What's special about the divergence of these particular clades compared to any other clades on the tree?

Ln 201 - Colubrids shouldn't be capitalised

Ln 206-207 - This time period is approximately coincident with the colonisation of the New World by vipers, which is potentially consistent with venom evolution being linked to ecological opportunity.

Ln 207-208 - You claim that "SVMP...showed an increase in evolutionary rates since the divergence of vipers", but Fig 3 seems to contradict this.

Ln 210 - "Ophiophagus hannah" should be 'Ophiophagus hannah'

Ln 219 - There should be an 'and', not a comma between "Brownian motion (BM)" and "Ornstein-Uhlenbeck (OU)".

Ln 220 - Why isn't Early burst a 'conventional' model? It's been a fairly standard part of the set of phenotypic evolution models for a pretty long time now.

Ln 292-294 - This again suggests that the most abundant (and often toxicologically dominant) toxins are likely the most important links to the ecology and evolution of the animals and their venom systems - they are probably more exposed to selection as a result.

Ln 298-299 - You didn't estimate ancestral character states (or at least don't present them in the manuscript), you estimate rate dynamics.

Ln 306-307 - Why not test for trait-dependent diversification rather than simply say 'it is tempting to speculate on this but there is a lack of evidence' (and, as above, you might want to discuss what evidence does exist for venom/toxin-associated diversification). You again say something similar on Ln 314-315 in that venom 'might not necessarily...show patterns of trait-dependent diversification', well, not necessarily but you could test for it.

Ln 312-313 - phenotypes don't lead to key innovations, phenotypes are the key innovations (the evolutionary consequences of which are what makes the difference to terminology)

Ln 329 - What do you mean by 'large-scale' diverse forms?

Ln 338 - It is very unlikely that snakes show mate choice based on venom composition, but that very direct mechanism is not necessary for venom phenotypes to "cause widespread character changes that are needed to establish reproductive isolation that leads to speciation". Indirect consequences would still ultimately be due to venom variation, for instance ecological separation (complete or partial) relating to diet specialism, foraging style, related habitat choice (for particular prey), differences in activity (better defended species may be less constrained by predation, be more active, and hence achieve a higher frequency of mating opportunities). As you allude to, little direct evidence for any of these types of mechanisms exist, but the important part is that neither direct venom-based mate choice nor widespread character changes (it could be one or two simple but important changes) are necessary.

Ln 346 - I think you mean 'adaptation' or 'adaptive radiation' instead of just "adaptive".

Ln 347-348 - Although you are correct that there is no way of "determining" (with absolute certainty) whether a clade with particular traits was more diverse in the past, there are plenty of ways to estimate this, so it's not quite so much of a lost cause as you suggest.

Ln 351-352 - Can you suggest plausible examples of such different conditions and processes that are relevant to venom evolution? Unless we have evidence of different processes we typically assume some degree of uniformitarianism (without which any historical science is impossible).

Ln 354 - Should "analysis" be 'analyses'?

Ln 355-356 - Just because you are necessarily basing your interpretations on the data you collected doesn't mean that your interpretations and findings are specific to your dataset only...at least I hope not as this would undermine the biological insights your study could give (or indeed any study ever published).

Ln 360-362 - I would have thought that the locations of rate shifts are far more relevant to your hypotheses than the number of rate shifts? Would your hypotheses be better supported if you found 6 vs 3 vs 1 shifts at random points in the phylogeny or particular rate shifts that occur at locations consistent with related evolutionary events you think are important?

Ln 371 - "misrepresented" should be 'misrepresented'

Ln 374 - you have doubled up the words "by the"

Ln 384-388 - I don't follow why a procedure applied to all species would only affect this one particular case.

Ln 403-405 - I agree that considering "the evolution of snake venom in terms of its impacts on speciation" would provide insight into venom as an influence in adaptive radiation, but as mentioned earlier, this has been tested in some other venomous groups (and also including, in combination with other tetrapods, in snakes). The existing studies focus on the presence of venom, rather than specific attributes, so there is more to be done here, but I think the lack of any discussion of this literature is a bit of an oversight (given you refer to the basic questions it aims to address at multiple points in your manuscript).

Ln 421-424 - How did you combine these two trees, and why did you choose to use two different phylogenies?

Ln 428-429 - something has gone a bit wrong with superscripts here.

Ln 430 - "conservatiove" should be 'conservative'

Ln 433 - By "significant from" do you mean 'significantly different from'?

Ln 434 - Convergence doesn't need to be capitalised.

Ln 443 - What do you mean by an "explosive EB" model? How is this different from any other EB model?

Ln 449-450 - I'm not sure what the sentence beginning with "one of the models" is saying, the wording is very confusing.

Ln 459 - When you say "weighted AIC was used as a measure of model fit" I'm not sure what you mean - do you mean AICc (which isn't mentioned elsewhere) or Akaike weights (aka model probabilities, which looks more similar to what is presented in ESM Table 1, but in this case I don't understand why the values in that Table don't sum to 1 in each row)? Particularly if you mean Akaike weights, these aren't a "measure of model fit" as such, but merely the probabilities of each model being the best in your model set (they say nothing about how good your model set is - none of them could fit well but you'll still get some better than others).

Ln 460-461 - The absolute values of AIC are meaningless for interpretation, and hence their ratio is as well, so I assume it's just unclear what you mean here but a model with twice the AIC of another model says nothing about the relative evidence each provides.

I hope the authors aren't too discouraged by these comments, and (as I said earlier) the manuscript is certainly in better shape now than the earlier version I reviewed. I do believe the basic idea and analyses here have something important to offer the field, but my comments are intended to help improve the presentation and interpretation of these results. I hope the authors find them useful.

Best wishes,
Kevin Arbuckle

Referee: 2

Comments to the Author(s)
Review of RSPB-2020-0122

General Comments

This is an interesting and significant investigation of the macroevolutionary consequences of possessing venom in snakes. The study infers rates and patterns of evolution in toxin gene expression through transcriptome techniques. The data are mined from published work and assembled and analyzed here to test various evolutionary pattern and process models for goodness of fit. In the end, the authors demonstrate that pulsed models, in which rates of evolution vary through time as if in response to shifting optima, fit the data better than random and early burst models. The core of the study is bracketed between an Introduction and a Discussion and Conclusion interpreting venom as a key innovation within a framework of adaptive radiation.

Overall, the paper is well written, genuinely compelling, and shows an understanding of many kinds of data from multiple subfields of evolutionary biology. There is something in this paper for everyone. The authors demonstrate an awareness of and acknowledge the limitations of their own data. They also are aware of the strengths of other kinds of data.

The overall rationale for the study, as presented here, is to test whether gene expression data conform to an early burst model of evolution better than phenotypic data. The authors present

the early burst model as the model expected of an adaptive radiation resulting from a key innovation that facilitates crossing an adaptive threshold and entering a new adaptive zone, followed by slower rates with the filling and subdivision of ecological niches. This view fits in with a recently proposed, and somewhat restrictive stance on key innovations. In the end, the authors found poor fit of venom toxins to the early burst model and better fit to pulsed models. This result implies that venom toxins do not just enable entering a new adaptive zone. Rather venom toxin continues to evolve with rapid pulsed episodes and to play a major role in the adaptations of snakes for capturing prey, perhaps enabling the coexistence of species, which supports higher standing diversity. In this respect, snake venom is not an anomaly, but rather is within a class of key innovations that promote further evolutionary change. Such adaptations are evolutionarily versatile in the sense of being highly evolvable and able to adapt quickly to changing conditions. I feel that the authors could make a more streamlined case in their Introduction if they took a more open-ended view of key innovations from the onset and discussed the evolutionary models (early burst and pulsed) that are consistent with scenarios on how different kinds of key innovations might operate. Doing so changes the focus from whether venom is a key innovation, which ultimately depends on one's definition of the term, to how venom has evolved and influenced the evolutionary radiation of venomous snakes.

Specific Comments

Title

The title summarizes the main results of this study about the tempo and mode of toxin evolution in snakes, but it does not fully convey what the paper is really about, which is whether snake venom is a key innovation and how it has influenced the radiation of snakes. Perhaps the title ought to reflect the full scope of the paper.

Abstract

Line 40. "are essential for prey capture" should perhaps be "facilitate prey capture". The latter wording is more active. Also, venom is essential only if a snake is restricted to a certain predation strategy (strike, follow, and wait) that requires it. Other snakes (boas and pythons) can also capture prey without venom, but with a different strategy and adaptations.

Line 51. Change "influence" to "influences". Insert "in" after "process" and "that" after "way". Also, "magic traits" is an odd phrase that requires explanation or at least a reference. Therefore, it may be better used in the main text than here in the abstract where it can neither be explained nor a reference cited.

Introduction

Line 60. "ecological speciation". Some authors (e.g., Mike Rosenzweig) have used this term to mean a kind of sympatric speciation, whereas other do not and treat it as a kind of allopatric speciation. The authors should probably explain what they themselves mean by using the term.

Lines 67-70. "Rather, the role of key innovations should be restricted to providing entry into novel ecological niches or adaptive zones, and studies should aim to identify specific shifts in tempo and mode of phenotypic evolution of the assumed key innovation (9,10)." If the authors wish to adopt this restricted definition of key innovation, or more precisely criteria for recognizing a key innovation, that is fine. However, if adopting this definition leads them to use the Early Burst model as the sine qua non of a key innovation, as they seem to do in the following paragraph, then they undermine their own argument. Ultimately, the authors find that the evolution of venom toxins in snakes is better explained by Pulsed models than by an Early Burst model. That does not surprise me because venom toxin seems like a evolutionarily versatile system as the authors describe it, a lot like the pharyngeal jaws of cichlid fishes in sense of facilitating further, rapid evolutionary change. Finding pulsed rate of evolution is what I would

expect of a key innovation that promotes further evolutionary change. The next thing that I would want to know is what are the tradeoffs. Are there any costs to having venom? How about of certain combinations of toxins? I would advise the authors to frame their argument about venom as a key innovation in comparison to other innovations that are versatile and highly evolvable. Such adaptations tend to be those with low costs to the organisms, often through relaxed trade-offs among functions.

Line 85. “rarely observed in data (14).” The authors should clarify what kind of data. Comparative data in extant clades, I should think. In the fossil record, the early burst pattern is more commonplace but by no means universal.

Lines 85-99 “This lack of empirical support might be because studies are overlooking the components of traits that actually produce phenotypic change; gene expression variation” and continuing to the end of the paragraph. I did not find this particular argument to be a convincing explanation for why few examples of early burst adaptive radiations occur among extant species. I do understand the authors’ main point that rates of evolution of genes and phenotypes may differ, and I understand the authors’ need to motivate their study around a specific question and a hypothesis. However, in adaptive evolution genes are only going to evolve as rapidly as the phenotypes to which they correspond are sorted by natural selection. One way in which genes might evolve faster is in drift among selectively neutral variants. But probably would not be the case here. Alternatively, and perhaps more of a problem, is in cases of highly developmentally canalized systems in which the same phenotype can develop from more than one, and perhaps multiple different gene expression patterns. This situation may be commonplace and could lead to higher rates of evolution in genes than in phenotype.

Lines 100-127. The problem of gene expression-phenotype mapping in complex traits versus in snake venom. The authors do make a good case here for the special status of venom because of its simplicity, in which different levels of gene expression result directly in different proportions among toxins in the venom “cocktail”. In other words, in snake venom one need not consider gene networks, morphogenetic pathways, and the like. The system sounds ideal. However, several thoughts occurred to me as I read this passage. What are the observed performance differences among toxin cocktails? What specific associations exist between toxin mixtures and either prey or environments? The idea that toxins do not interact with other is presented rather definitively and with citation of just one supporting reference. How was this inference made? Although the authors present toxins within venom as isolated adaptations without interactions, venom requires a delivery system (glands and fangs) as they later mention briefly, a specific hunting strategy (behavior and supporting morphology, including sensations), and either immunity from the effects of one’s own toxins, some barrier sequestering toxins, or at least storage of toxins in an inactive state until deployed. How is all this potential complexity managed? One could imagine far-reaching consequences of having venom or of variation among venom cocktails on the phenotypes of snakes. Is venom really such an isolated system?

Lines 133-135. “If snake venoms are key innovations we would expect to see shifts in their evolutionary rates in response to changes in optima.” How are optima identified? Ideally, this would be by reference to evidence external to the snakes themselves, such as the timing of environmental changes from the paleoclimatic record, biogeographic dispersal events into new regions, or time of contact with novel prey. There is a certain circularity to inferring the existence of optima from the evolutionary rate dynamics of snake venom alone. Is there external evidence too?

Results

Lines 232-235. “Jump models have highest weighted AIC scores and are a better fit to snake venom gene expression data as compared to conventional incremental models of evolution (Table 1; electronic supplementary material).” Are the jump models grouped together in Table 1 under the heading “Pulsed”? If yes, then please state so. In general, yes, the “Pulsed” models all fit

better than the random (BM and OU) and early burst (EB) models. Do the pulsed models fit significantly better? Also, can the “pulsed” models distinguish oscillation between stasis and random walk, on the one hand, from sporadic pulses of directional change, on the other hand? In order to relate these models to those traditionally used to study evolutionary rates in paleontology (stasis, random walk, and directional evolution), it would be beneficial if the authors would explain the correspondences between these models and traditional ones.

Lines 273-281. Optima, prey diversity, and environment. So far, the existence of optima has only been inferred in the authors’ data from the good fit of the data to pulsed (Lévy process) models that assume optima. The authors could make a stronger argument if they could be more specific about what the optima are and how venom has evolved specifically to allow snakes to adapt to these shifting optima. If specific associations are lacking, then at the very least they could explain how prey diversity and environment could in theory co-define optima as the authors claim. For example, what toxin combinations are better suited to a diverse range of prey, and which are better suited to one or a few prey types? Do venomous snakes conform to the specialist/generalist dichotomy? Are there tradeoffs between toxin efficacy on one kind of prey and the number of prey types on which it can be used? How does environment factor in? Through general levels of metabolic and behavioral activity expected of an ectotherm? Or in more specific ways through prey availability, overall or seasonally? Right now, these optima are just a little too vague to be totally satisfying.

Lines 346-349. “Nearly all studies of adaptive [change to “adaptation”] focus on traits and processes in extant species, [and] this is a major disadvantage since there is no way of representing extinct taxa and thus no way of determining whether a clade with specific innovations was more species[-]rich in the past (77,78).” Well said, and very refreshing to see this statement here. Please note minor edits in brackets.

Author's Response to Decision Letter for (RSPB-2020-0122.R0)

See Appendices A & B.

RSPB-2020-0613.R0

Review form: Reviewer 1 (Kevin Arbuckle)

Recommendation

Accept as is

Scientific importance: Is the manuscript an original and important contribution to its field?

Excellent

General interest: Is the paper of sufficient general interest?

Good

Quality of the paper: Is the overall quality of the paper suitable?

Excellent

Is the length of the paper justified?

Yes

Should the paper be seen by a specialist statistical reviewer?

No

Do you have any concerns about statistical analyses in this paper? If so, please specify them explicitly in your report.

No

It is a condition of publication that authors make their supporting data, code and materials available - either as supplementary material or hosted in an external repository. Please rate, if applicable, the supporting data on the following criteria.

Is it accessible?

Yes

Is it clear?

Yes

Is it adequate?

Yes

Do you have any ethical concerns with this paper?

No

Comments to the Author

Dear authors,

I am delighted to see the attention given to my previous comments and have almost nothing more to add.

One very minor remaining point which has been highlighted but not corrected is on lines 465-466 the phrase "if its AIC is at least twice as high as its competing model" still persists. As written this is incorrect since AIC and Akaike weights are not synonymous – the ratio of AIC (as written) is meaningless, but the ratio of Akaike weights is indeed useful. Nevertheless, a tweak as minor as changing "AIC" to "Akaike weights" can be addressed at the proofing stage, and I have no further hesitation to recommend this manuscript for publication.

I would only like to congratulate the authors on an interesting and well-conducted study that adds new perspectives to the emerging field of venom macroevolution.

Best wishes,
Kevin Arbuckle

Decision letter (RSPB-2020-0613.R0)

27-Mar-2020

Dear Mr Barua

I am pleased to inform you that your manuscript RSPB-2020-0613 entitled "Toxin expression in snake venom evolves rapidly with constant shifts in evolutionary rates" has been accepted for publication in Proceedings B.

The referee(s) have recommended publication, but also suggest some minor revisions to your manuscript. Therefore, I invite you to respond to the referee(s)' comments and revise your manuscript. Because the schedule for publication is very tight, it is a condition of publication that you submit the revised version of your manuscript within 7 days. If you do not think you will be able to meet this date please let us know.

- DNA sequences: Genbank accessions F234391-F234402

- Phylogenetic data: TreeBASE accession number S9123
- Final DNA sequence assembly uploaded as online supplemental material
- Climate data and MaxEnt input files: Dryad doi:10.5521/dryad.12311

[http://datadryad.org/submit?journalID=RSPB&manu=\(Document not available\)](http://datadryad.org/submit?journalID=RSPB&manu=(Document%20not%20available)) which will take you to your unique entry in the Dryad repository. If you have already submitted your data to dryad you can make any necessary revisions to your dataset by following the above link. Please see <https://royalsociety.org/journals/ethics-policies/data-sharing-mining/> for more details.

Sincerely,
Dr Sasha Dall
mailto:proceedingsb@royalsociety.org

Associate Editor
Board Member
Comments to Author:
Dear Authors,

I am happy to tell that one of the original reviewers have looked over the revised version of the manuscript and has only one minor point to correct (which does not require further reviewing). I agree that you have done good work with the revision and I have absolutely no problem to bumping this up to the deciding editor as manuscript recommended for publication in Proc B.

Best wishes
Juha Merilä

Reviewer(s)' Comments to Author:

Referee: 1

Comments to the Author(s).
Dear authors,

I am delighted to see the attention given to my previous comments and have almost nothing more to add.

One very minor remaining point which has been highlighted but not corrected is on lines 465-466 the phrase "if its AIC is at least twice as high as its competing model" still persists. As written this is incorrect since AIC and Akaike weights as they are not synonymous – the ratio of AIC (as written) is meaningless, but the ratio of Akaike weights is indeed useful. Nevertheless, a tweak as minor as changing "AIC" to "Akaike weights" can be addressed at the proofing stage, and I have no further hesitation to recommend this manuscript for publication.

I would only like to congratulate the authors on an interesting and well-conducted study that adds new perspectives to the emerging field of venom macroevolution.

Best wishes,
Kevin Arbuckle

Author's Response to Decision Letter for (RSPB-2020-0613.R0)

See Appendices C & D.

Decision letter (RSPB-2020-0613.R1)

30-Mar-2020

Dear Mr Barua

I am pleased to inform you that your manuscript entitled "Toxin expression in snake venom evolves rapidly with constant shifts in evolutionary rates" has been accepted for publication in Proceedings B.

Open Access

Paper charges

Sincerely,
Editor, Proceedings B
mailto: proceedingsb@royalsociety.org

Appendix A

Reviewer(s)' Comments to Author:

Referee: 1

Comments to the Author(s)

Firstly, I should disclose that I have already reviewed this manuscript elsewhere on a prior submission (to my knowledge this is no longer being considered there so there is no conflict), and I am glad to see that substantial changes have been made which, on the whole I think have improved the manuscript. Nevertheless, I still have outstanding concerns/comments here, partly issues that remain from my earlier review and partly new comments introduced in the current version. I think this manuscript is well placed to make an impact in the field if these comments are addressed, and so I list them below (in roughly chronological order) in the hope that they are useful to the authors.

Ln 41-42 - I think you need to clarify your comment that there have been 'no quantitative analyses' testing whether venom is a key innovation. There have been tests of the hypothesis that venom is associated with higher diversification rates, which is often regarded as a major component of key innovations and, more recently, has been regarded as a non-essential but (nevertheless) a common outcome of key innovations (and by many definitions is the 'radiation' part of adaptive radiation).

Thank you for the clarification. We modified the abstract to focus on rate dynamics instead:

Ln 40-42 - Venoms evolve rapidly, are essential for prey capture, and are widely believed to be key innovations leading to adaptive radiation. However, few studies have estimated their evolutionary rate dynamics.

Ln 51 - I'm not aware of any cases where someone has proposed that venom acts as a 'magic trait', merely as a key innovation.

Removed 'magic traits' replaced with:

In 51-52 - Therefore, the extent to which venom directly influences the diversification process is still a matter of contention.

Ln 108-111 and 119-120 - I'm not convinced by the relevance of your assertion that "highly tissue specific genes would also likely reduce significant pleiotropic constraints and cross-phenotype associations helping to discern the unique trajectories of individual genes". While tissue-specific expression would certainly reduce such interactions across tissues, many venom toxins are widely suspected or known to have lots of pleiotropic functional effects and synergistic or antagonistic interactions with other venom components. In other words, as a highly specialised and multidimensional (complexity of different interacting toxins) system I

don't think that venom molecular evolution can be expected to typically have low rates of pleiotropy or molecular interactions.

We agree with this point and have modified the text accordingly:

Ln 116-122 - Venom toxins can have both agonistic and antagonistic interactions between other toxin components, but how they influence other traits outside the venom system is unclear. On one hand venoms are integrated systems with different toxins acting in concert to immobilise prey (Fry et al. 2006). On the other hand, whether this mode of action introduces an evolutionary constraint is less clear, since there is little phylogenetic covariance between components (Barua and Mikheyev 2019).

Ln 128-129 - Depending on your definition of key innovation you could perhaps argue that phenotypic differentiation driven by ecology has not been studied well in snakes (although you could easily consider some of the work on diet and venom evolution to fall into this bracket), however, when talking about "the extensive radiation of snakes", which strongly implied lineage diversification, it is untrue to claim that "no studies have tested this hypothesis". In fact, I find it surprising that none of the work linking venom evolution to lineage diversification rates appears in this paper given its clear relevance at various points. In the interest of full disclosure, some of this work is my own (e.g. Harris and Arbuckle, 2016, Toxins 8:193) but other work is not (e.g. Liu et al., 2018, Mol Phylogenet Evol 125:138-146).

Thank you for the suggestion. We have modified the text to include the following:

Ln 130-151 - The idea that venom is a key innovation and that it underlies the extensive radiation of snakes is pervasive in the literature (2,43–50). Yet, few studies have examined long-term changes in evolutionary rates of venom gene expression in snakes. There are numerous studies that have examined the role of venom in lineage diversification in other taxa (Harris and Arbuckle 2016; Arbuckle and Speed 2015; Liu et al. 2018; Blanchard and Moreau 2017). In blenny fish the presence of a venom system in the form of a buccal gland and fang is associated with higher rates of diversification (Liu et al. 2018). In tetrapods, evolution of venoms and poisons are typically associated with an increase in diversification rates (except in amphibians) (Harris and Arbuckle 2016). There is also a substantial amount of literature suggesting the role of diet in lineage diversification (Rainford and Mayhew 2015; Burin et al. 2016; Rojas et al. 2018). Since snakes use venom primarily for prey procurement, alterations in venom and diet could have an effect on diversification in venomous snakes.

Key innovations however have more features than just causing lineage diversification. Key innovations contribute to expansion of ecological ranges, they represent optimal adaptations, and usually undergo changes in evolutionary rates to fill morphospace (Givnish 2015). Restricting the role of key innovations to just diversification ignores these features, and removes emphasis on evolution of the key innovation itself (Givnish 2015; Rabosky 2017). In this study, we specifically focus on the evolution of snake venom. We use a comparative dataset of snake venom gene

expression to identify shifts in macroevolutionary rates, which are characteristic of key innovations (Rabosky 2017). To further characterize patterns of venom evolution, we estimated long-term changes in evolutionary rates of venom gene expression, and also fit the data to several trait-evolution models. Our results revealed that toxin expression in snake venom evolves very rapidly, and has experienced numerous shifts in evolutionary rates over the past 60 million years.

Ln 135 - optima for what?

We have removed this line as it is confusing.

Ln 145-147 - How did you deal with cases of multiple studies of the transcriptomes for a single species?

For consistency, we use the previously collected and carefully curated dataset from (Barua and Mikheyev 2019). Since this data was collected in 2017, it doesn't include studies published after that.

Ln 159 - Fig 1 seems to be missing. I am unclear from the wording here whether you are referring to Fig 1 within the Supplementary Material or both Fig 1 and Supplementary Material, but in either case there is no Fig 1 in your manuscript (numbering starts at Fig 2).

Should be in the electronic supplementary material. We will double-check next time if it was uploaded properly.

Ln 163-165 - What is the relevance to your study of age of the clade in your study? It seems an odd detail to add with no explanation of its relevance.

Since we use a time calibrated tree we decided to mention the estimate of the root, which some readers find useful as it gives them a perspective of time. We have not made any modifications to the text in regard to this.

Ln 180 - "toxins families" should be 'toxin families'

Ln -194 - Have fixed in the manuscript.

Ln 183-192 - There are a few points of interpretation of your data that I'm not sure you have considered (at most having a couple of partially related comments later in the manuscript for some of these points but not fully incorporated or explored). Of the toxins with the best evidence of 'early burst' patterns (which are potentially indicative of adaptive radiation) two important toxins stand out - SVMPs and TFTxs. Specifically, these are often the most abundant toxins, are often toxicologically dominant ones, and the 'early burst' patterns are largely present in vipers (for SVMPs, colubrids also show this pattern) and elapids (for TFTxs, also stated on Ln 211), which also corresponds to the main toxin classes in each of these lineages. The

alternative lineages for these toxin patterns may be evidence for trade-offs (as shown previously by the authors - in practice if not necessarily these toxin classes appear as 'alternative' venom composition types).

Thank you. Following this suggestion we decided to rethink the rate through time analysis. We decided not to use rate through time plots to infer early burst, considering that no study has used it in that manner, and as we now feel it isn't quite capable of characterising early burst evolution well. Instead, we focus specifically on observing rate patterns in venom expression evolution in the three families. We test for early burst evolution using the modelling approaches discussed in the next sections, which provide a more direct comparison of various alternative hypotheses. We have modified text accordingly:

Ln 208-212 - Under adaptive radiation hypothesis, ecomorphological rates should transition from rapid rates to slow, equilibrium rates as ecological niche gets filled (Slater et al. 2010). To identify these patterns we estimated the rates of toxin expression evolution of each toxin family after the split of the three families. Our estimates of 'rate through time' revealed that toxin families show unique evolutionary rates and rate dynamics in each snake family.

Ln 228-231 - SVMP, TFTx show an interesting pattern where they seem to represent alternate venom types. SVMP has high rates, and is dominant in vipers (and to an extent in colubrids) while TFTx has higher rates, and is predominant in elapids. The alternate lineage of these toxins could be evidence of tradeoffs, a pattern that we previously observed (Barua and Mikheyev 2019).

Importantly, the time-scale of toxin origins in relation to your clade under study is important for interpreting patterns of evolutionary rate dynamics. In your results, some toxins start off with very low rates and show patterns consistent with early burst style dynamics (e.g. C-type Lectins and SPs) or maintain low rates across most of the tree (e.g. KSPIs). As these are predicted to pre-date the root of your tree they may have already experienced an early burst before slowing prior to the earliest point of your study. Consistent with this idea is that the end of the early burst style patterns of SVMP and 3FTxs have estimated rates comparable to the root estimates of other toxins. For some of the patterns you are looking for it is important to have the origin of the trait within the phylogeny and this important caveat is neither stated nor discussed (the closest is on Ln 296-298 but this failed to acknowledge that almost all the toxin families here pre-date you clade, and most of them originated on one of two branches).

Thank you for raising this important point. You are absolutely correct that ideally the traits should have originated within the time points of our tree, and that it is perfectly plausible that these toxins may have experienced the early burst pattern before the earliest point in our tree. However, we are more interested in their recent role in evolution. All of the lineage divergence in our tree occurred well after the root. Therefore, if the traits were already present before the diversification, they might not have contributed substantially to the diversification of extant lineages (which is the predominant belief in the field). Rather than recruitment, if changes in

expression level of the trait contributed to diversification, we would expect to see the patterns that we observe in toxins like C-type Lectins and SPs.

We have expanded the discussion to to incorporate this likely caveat:

Ln 312-323 - Rate dynamics and age also tend to be related, with older toxins experiencing higher rates in the past followed by reduction in modern lineages (Fig 2). It should be noted that the probable origin of most of the toxin families in our study pre-dates the root of our tree (Fry et al. 2006). Typically, if a trait is responsible for lineage diversification, its origin should be at a point within the tree around the time of major branching events. However, we can only examine what happens to venom evolution after the most recent common ancestor of extant snakes. Some of the oldest toxins to comprise the venom: SVMP, TFTx, and CRISP (Fry et al. 2008) all showed a larger distribution of high evolutionary rates near the root than the tips (Fig 2). This could be because the major toxin families were likely present in the ancestral venom and experienced a uniform reduction in evolutionary rates as lineages diversified. These toxins that pre-date the root likely allowed ancestral snake lineages to realize their ecological potential which lead to niche specification, which in turn lead to a slowdown in evolutionary rates in their descendants.

Ln 195-197 - How are you defining "major lineages" in the context of 'expectations that rates of trait evolution should be high immediately after they diverge'? Why is the split of the three venomous families an important evolutionary point for your study (note also that you should check if 'Colubridae' is actually the family you mean as this 'traditional wastebasket taxon' has been split into many families)? What's special about the divergence of these particular clades compared to any other clades on the tree?

For this we actually use a convention laid out in the BAMM tutorial under the 'rate through time' section (<http://bamm-project.org/postprocess.html#rate-through-time-analysis>). This way of computing rates from a specific point of interest in the tree is common and widely used (Rabosky et al. 2014; Fernández-Mazuecos et al. 2019; Folk et al. 2019). We selected these nodes specifically because they are the points of origin of the three major venomous snake families in our tree.

We agree that classifications of 'Colubridae' are sometimes unclear. For our study we broadly classify the non viper and non-elapid lineage as colubrids. Since we used the same tree and same classification in our previous study, we decided to do the same for consistency.

Ln 201 - Colubrids shouldn't be capitalised

Ln 214 - Have fixed the error.

Ln 206-207 - This time period is approximately coincident with the colonisation of the New World by vipers, which is potentially consistent with venom evolution being linked to ecological opportunity.

We modified the text to include this:

Ln 218-223 - In vipers, toxin families generally showed an increase in evolutionary rates, with a majority of them occurring at around the 20-million-year mark, just after the diversification of the major viperid lineages, which is potentially consistent with venom evolution being linked to ecological opportunity.

Ln 207-208 - You claim that "SVMP...showed an increase in evolutionary rates since the divergence of vipers", but Fig 3 seems to contradict this.

Correct. It's my mistake. Have deleted the sentence.

Ln 210 - "Ohiophagus hannah" should be 'Ophiophagus hannah'

Ln 225 - Done.

Ln 219 - There should be an 'and', not a comma between "Brownian motion (BM)" and "Ornstein-Uhlenbeck (OU)".

Ln 237-238 - Yes. Corrected.

Ln 220 - Why isn't Early burst a 'conventional' model? It's been a fairly standard part of the set of phenotypic evolution models for a pretty long time now.

Ln 237-238 - Agreed. We have clubbed them together.

Ln 292-294 - This again suggests that the most abundant (and often toxicologically dominant) toxins are likely the most important links to the ecology and evolution of the animals and their venom systems - they are probably more exposed to selection as a result.

Agreed. We have modified the text to include this suggestion:

Ln 307-311- Abundant toxin families along with ePLA2 and SVSP showed a higher net rate of evolution of gene expression (Fig. 3), a trend also observed in sequence data (Aird et al. 2015, 2017). This suggests that the most abundant (and often toxicologically dominant) toxins have the strongest links to the ecology and evolution of snakes and their venom systems - they are also probably more exposed to selection as a result.

Ln 298-299 - You didn't estimate ancestral character states (or at least don't present them in the manuscript), you estimate rate dynamics.

Correct, we have removed the sentence.

Ln 306-307 - Why not test for trait-dependent diversification rather than simply say 'it is tempting to speculate on this but there is a lack of evidence' (and, as above, you might want to discuss what evidence does exist for venom/toxin-associated diversification). You again say something similar on Ln 314-315 in that venom 'might not necessarily...show patterns of trait-dependent diversification', well, not necessarily but you could test for it.

Testing trait dependent diversification for continuous traits required a large amount of data which is frankly not available in our case. Also, along the lines of our narrative, we do not want to test for trait dependent diversification because we believe it's a restricted perspective of key innovations (as mentioned before). The predominant belief that we are trying to challenge in that key innovations provide ecological opportunity, but not necessarily lead to lineage diversification. When looking at patterns of trait evolution many of the dynamics are attributed to causing species divergence. We want the reader to appreciate that the patterns we observe are characteristic of key innovations, but these should not be considered as evidence for trait dependent diversification. We want to stress on the fact that species divergence does not happen through an isolated mechanism, but requires many specific conditions, which may be difficult to meet (as mentioned in the discussion). Therefore, while it is easy to interpret our observations as evidence of species diversification, one should not jump the gun without considering all the possible lines of evidence. A similar argument has been made in this recent study, with a much larger sample size, from which we can draw parallels(Phuong et al. 2019).

Ln 312-313 - phenotypes don't lead to key innovations, phenotypes are the key innovations (the evolutionary consequences of which are what makes the difference to terminology)

Ln330-331 - Apologies for the misconception. Have clarified in the text.

Ln 329 - What do you mean by 'large-scale' diverse forms?

We mean that evolution of modular traits would likely not be associated with or cause evolution of other traits. For example, venom evolution would not lead to morphological changes outside the venom system, like developing appendages.

Ln 338 - It is very unlikely that snakes show mate choice based on venom composition, but that very direct mechanism is not necessary for venom phenotypes to "cause widespread character changes that are needed to establish reproductive isolation that leads to speciation". Indirect consequences would still ultimately be due to venom variation, for instance ecological separation (complete or partial) relating to diet specialism, foraging style, related habitat choice (for particular prey), differences in activity (better defended species may be less constrained by predation, be more active, and hence achieve a higher frequency of mating opportunities). As you allude to, little direct evidence for any of these types of mechanisms exist, but the important part if that neither direct venom-based mate choice nor widespread character changes (it could be one or two simple but important changes) are necessary.

Thank you for this insight. We have included this in our text as it makes our argument more complete.

Ln358-363 - It is not known if snakes have a similar kind of mating preference based on venom composition. For example, would a female prefer a male with more similar or dissimilar venom composition for mating? Venom could however, lead to indirect ecological consequences in terms of foraging style, habitat choice, temporal differences in activity. But speciation requires a level of reproductive isolation, how this is achieved either directly or indirectly by changes to the venom is not obvious.

Ln 346 - I think you mean 'adaptation' or 'adaptive radiation' instead of just "adaptive".

Ln369 - Thank you. We have fixed it.

Ln 347-348 - Although you are correct that there is no way of "determining" (with absolute certainty) whether a clade with particular traits was more diverse in the past, there are plenty of ways to estimate this, so it's not quite so much of a lost cause as you suggest.

Sure, we'll try and tone down the cynicism there :-)

Ln 351-352 - Can you suggest plausible examples of such different conditions and processes that are relevant to venom evolution? Unless we have evidence of different processes we typically assume some degree of uniformitarianism (without which any historical science is impossible).

We added this line as a general caveat to these kinds of studies. We included the following in the text:

Ln373-380 - While most studies provide a microevolutionary perspective, extrapolating from processes that operate in the present day to what happened early in a clade's history is difficult; because conditions were different in the past, different processes may have been at work, or may have produced different outcomes. Perhaps in the past there were venomous snakes with venom compositions specific to the past environment. In response to any changes in this environment snakes could have evolved venom composition starkly different from the ones we see today. There might also have been venomous snake lineages in the past that became extinct leaving a whole history of venom composition unexplored.

Ln 354 - Should "analysis" by 'analyses'?

See next comment.

Ln 355-356 - Just because you are necessarily basing your interpretations on the data you collected doesn't mean that your interpretations and findings are specific to your dataset

only...at least I hope not as this would undermine the biological insights your study could give (or indeed any study ever published).

Thank you for this input. We agree and have removed the sentence. We realize that it didn't read well, and didn't convey our intended message.

Ln 360-362 - I would have thought that the locations of rate shifts are far more relevant to your hypotheses than the number of rate shifts? Would your hypotheses be better supported if you found 6 vs 3 vs 1 shifts at random points in the phylogeny or particular rate shifts that occur at locations consistent with related evolutionary events you think are important?

While the location of the rate shifts are important, the way BAMM works, it's advised not to stress specifically on the location of the rate shifts:

"In the BAMM framework, there is no single set of independent rate shifts waiting to be identified. Rather, BAMM identifies configurations of rate shifts - sets of shifts that are sampled together -

The model assumes that a relatively small number of discrete shift events can explain the data (sort of.... it is more that the marginal likelihood of any particular model is implicitly penalized by the addition of more parameters).It is possible that many shifts in evolutionary dynamics change in a discrete fashion (e.g., the classic "key innovation" scenario), but it is also possible that major changes in dynamics occur through a number of sequential changes in some general region of a tree"

<http://bamm-project.org/rateshifts.html>

However, if rate shifts occur in unexpected, spurious locations (like they do for ePLA2 and TFTx in Crotalus), they have to be explained.

Ln 371 - "missrepresented" should be 'misrepresented'

Fixed.

We have also added the caveat sections to the supplementary as it allows a greater exploration of caveats and their solutions without being limited by the word limit of the journal. We have mentioned this in the relevant methods section of the manuscript.

Ln 374 - you have doubled up the words "by the"

Fixed.

Ln 384-388 - I don't follow why a procedure applied to all species would only affect this one particular case.

This is because these two Crotalus species are very closely related and have very different venoms from one each other (one has more structural zeros than the other). The normalization (scaled around the mean) is done within species and this inflates the differences between them. Like mentioned this is appropriate for toxins that are actually present in both the snakes. But if a toxin is absent in both the snakes, its normalized value would depend on the value of the other toxins that are present. This creates an artificial difference between both the species for the missing toxin.

If the normalization was causing spurious differences throughout the dataset we would see it in other closely related species as well; Protobothrops, which like Crotalus do not have ePLA2, or Naja, and Micrurus, which do not have vPLA2. Since we do not see spurious shifts in these lineages, we can be fairly certain that the normalization only had an effect in the two Crotalus species.

Ln 403-405 - I agree that considering "the evolution of snake venom in terms of its impacts on speciation" would provide insight into venom as an influence in adaptive radiation, but as mentioned earlier, this has been tested in some other venomous groups (and also including, in combination with other tetrapods, in snakes). The existing studies focus on the presence of venom, rather than specific attributes, so there is more to be done here, but I think the lack of any discussion of this literature is a bit of an oversight (given you refer to the basic questions it aims to address at multiple points in your manuscript).

Agreed. As mentioned before, we have included references to several papers that talk about this.

Ln 421-424 - How did you combine these two trees, and why did you choose to use two different phylogenies?

We used an already published time-calibrated phylogeny from (Zheng and Wiens 2016). This phylogeny is one of the best ones in literature and is perfect for large comparative analysis.

Ln 428-429 - something has gone a bit wrong with superscripts here.

Ln431-432 - Thank you. It is fixed.

Ln 430 - "conservatiove" should be 'conservative'

Ln433 - Thank you. It is fixed.

Ln 433 - By "significant from" do you mean 'significantly different from'?

Ln 436 - Yes, thank you.

Ln 434 - Convergence doesn't need to be capitalised.

Ln437 - Thank you. It is fixed..

Ln 443 - What do you mean by an "explosive EB" model? How is this different from any other EB model?

Ln449 - It is not different,we have fixed this.

Ln 449-450 - I'm not sure what the sentence beginning with "one of the models" is saying, the wording is very confusing.

Apologies. We have modified the sentence:

Ln 455-457 - We model stasis followed by rapid adaptation using a compound Poisson process. This is the Jump Normal (JN) process, which assumes jump sizes are drawn from a normal distribution.

Ln 459 - When you say "weighted AIC was used as a measure of model fit" I'm not sure what you mean - do you mean AICc (which isn't mentioned elsewhere) or Akaike weights (aka model probabilities, which looks more similar to what is presented in ESM Table 1, but in this case I don't understand why the values in that Table don't sum to 1 in each row)? Particularly if you mean Akaike weights, these aren't a "measure of model fit" as such, but merely the probabilities of each model being the best in your model set (they say nothing about how good your model set is - none of them could fit well but you'll still get some better than others).

Thank you for the clarification. Yes indeed, AIC weights do not represent model fit, but they are used to select the best performing model(Wagenmakers and Farrell 2004). We use the AIC weight to determine which model best suits our data. The values in our table represent AIC weights for each of the 9 models we tested (BM, OU, EB, and 6 pulsed models). In all cases the pulsed models were favoured as compared to the non pulsed models. However, each pulsed model had very similar weights, which make it difficult to determine which pulsed model is better. For that reason we club them together and report the highest AIC weight.

Ln 460-461 - The absolute values of AIC are meaningless for interpretation, and hence their ratio is as well, so I assume it's just unclear what you mean here but a model with twice the AIC of another model says nothing about the relative evidence each provides.

Ratio of AIC weights are used to determine the best model among a group of models(Wagenmakers and Farrell 2004). Our use of a criterion for a better model follows the approach used in (Landis and Schraiber 2017).

.....

I hope the authors aren't too discouraged by these comments, and (as I said earlier) the manuscript is certainly in better shape now than the earlier version I reviewed. I do believe the basic idea and analyses here have something important to offer the field, but my comments are intended to help improve the presentation and interpretation of these results. I hope the authors find them useful.

We appreciate the reviewer's continued investment into the quality of this study.

Best wishes,
Kevin Arbuckle

Refs

- Aird, Steven D., Shikha Aggarwal, Alejandro Villar-Briones, Mandy Man-Ying Tin, Kouki Terada, and Alexander S. Mikheyev. 2015. "Snake Venoms Are Integrated Systems, but Abundant Venom Proteins Evolve More Rapidly." *BMC Genomics* 16 (August): 647.
- Aird, Steven D., Jigyasa Arora, Agneesh Barua, Lijun Qiu, Kouki Terada, and Alexander S. Mikheyev. 2017. "Population Genomic Analysis of a Pitviper Reveals Microevolutionary Forces Underlying Venom Chemistry." *Genome Biology and Evolution* 9 (10): 2640–49.
- Fernández-Mazuecos, Mario, José Luis Blanco-Pastor, Ana Juan, Pau Carnicero, Alan Forrest, Marisa Alarcón, Pablo Vargas, and Beverley J. Glover. 2019. "Macroevolutionary Dynamics of Nectar Spurs, a Key Evolutionary Innovation." *The New Phytologist* 222 (2): 1123–38.
- Folk, Ryan A., Rebecca L. Stubbs, Mark E. Mort, Nico Cellinese, Julie M. Allen, Pamela S. Soltis, Douglas E. Soltis, and Robert P. Guralnick. 2019. "Rates of Niche and Phenotype Evolution Lag behind Diversification in a Temperate Radiation." *Proceedings of the National Academy of Sciences of the United States of America* 116 (22): 10874–82.
- Fry, Bryan G., Holger Scheib, Louise van der Weerd, Bruce Young, Judith McNaughtan, S. F. Ryan Ramjan, Nicolas Vidal, Robert E. Poelmann, and Janette A. Norman. 2008. "Evolution of an Arsenal: Structural and Functional Diversification of the Venom System in the Advanced Snakes (Caenophidia)." *Molecular & Cellular Proteomics: MCP* 7 (2): 215–46.
- Givnish, Thomas J. 2015. "Adaptive Radiation versus 'Radiation' and 'Explosive Diversification': Why Conceptual Distinctions Are Fundamental to Understanding Evolution." *The New Phytologist* 207 (2): 297–303.
- Harris, Richard J., and Kevin Arbuckle. 2016. "Tempo and Mode of the Evolution of Venom and Poison in Tetrapods." *Toxins* 8 (7). <https://doi.org/10.3390/toxins8070193>.
- Landis, Michael J., and Joshua G. Schraiber. 2017. "Pulsed Evolution Shaped Modern Vertebrate Body Sizes." *Proceedings of the National Academy of Sciences of the United States of America* 114 (50): 13224–29.
- Liu, Shang-Yin Vanson, Bruno Frédérick, Sébastien Lavoué, Jonathan Chang, Mark V.

- Erdmann, Gusti Ngurah Mahardika, and Paul H. Barber. 2018. "Buccal Venom Gland Associates with Increased of Diversification Rate in the Fang Blenny Fish *Meiacanthus* (Blenniidae; Teleostei)." *Molecular Phylogenetics and Evolution* 125 (August): 138–46.
- Phuong, Mark A., Michael E. Alfaro, Gusti N. Mahardika, Ristiyanti M. Marwoto, Romanus Edy Prabowo, Thomas von Rintelen, Philipp W. H. Vogt, Jonathan R. Hendricks, and Nicolas Puillandre. 2019. "Lack of Signal for the Impact of Conotoxin Gene Diversity on Speciation Rates in Cone Snails." *Systematic Biology* 68 (5): 781–96.
- Rabosky, Daniel L. 2017. "Phylogenetic Tests for Evolutionary Innovation: The Problematic Link between Key Innovations and Exceptional Diversification." *Philosophical Transactions of the Royal Society of London. Series B, Biological Sciences* 372 (1735). <https://doi.org/10.1098/rstb.2016.0417>.
- Rabosky, Daniel L., Stephen C. Donnellan, Michael Grundler, and Irby J. Lovette. 2014. "Analysis and Visualization of Complex Macroevolutionary Dynamics: An Example from Australian Scincid Lizards." *Systematic Biology* 63 (4): 610–27.
- Slater, Graham J., Samantha A. Price, Francesco Santini, and Michael E. Alfaro. 2010. "Diversity versus Disparity and the Radiation of Modern Cetaceans." *Proceedings. Biological Sciences / The Royal Society* 277 (1697): 3097–3104.
- Wagenmakers, Eric-Jan, and Simon Farrell. 2004. "AIC Model Selection Using Akaike Weights." *Psychonomic Bulletin & Review* 11 (1): 192–96.
- Zheng, Yuchi, and John J. Wiens. 2016. "Combining Phylogenomic and Supermatrix Approaches, and a Time-Calibrated Phylogeny for Squamate Reptiles (lizards and Snakes) Based on 52 Genes and 4162 Species." *Molecular Phylogenetics and Evolution* 94 (Pt B): 537–47.

Appendix B

Referee: 2

Comments to the Author(s)
Review of RSPB-2020-0122

General Comments

This is an interesting and significant investigation of the macroevolutionary consequences of possessing venom in snakes. The study infers rates and patterns of evolution in toxin gene expression through transcriptome techniques. The data are mined from published work and assembled and analyzed here to test various evolutionary pattern and process models for goodness of fit. In the end, the authors demonstrate that pulsed models, in which rates of evolution vary through time as if in response to shifting optima, fit the data better than random and early burst models. The core of the study is bracketed between an Introduction and a Discussion and Conclusion interpreting venom as a key innovation within a framework of adaptive radiation.

Overall, the paper is well written, genuinely compelling, and shows an understanding of many kinds of data from multiple subfields of evolutionary biology. There is something in this paper for everyone. The authors demonstrate an awareness of and acknowledge the limitations of their own data. They also are aware of the strengths of other kinds of data.

The overall rationale for the study, as presented here, is to test whether gene expression data conform to an early burst model of evolution better than phenotypic data. The authors present the early burst model as the model expected of an adaptive radiation resulting from a key innovation that facilitates crossing an adaptive threshold and entering a new adaptive zone, followed by slower rates with the filling and subdivision of ecological niches. This view fits in with a recently proposed, and somewhat restrictive stance on key innovations. In the end, the authors found poor fit of venom toxins to the early burst model and better fit to pulsed models. This result implies that venom toxins do not just enable entering a new adaptive zone. Rather venom toxin continues to evolve with rapid pulsed episodes and to play a major role in the adaptations of snakes for capturing prey, perhaps enabling the coexistence of species, which supports higher standing diversity. In this respect, snake venom is not an anomaly, but rather is within a class of key innovations that promote further evolutionary change. Such adaptations are evolutionarily versatile in the sense of being highly evolvable and able to adapt quickly to changing conditions.

I feel that the authors could make a more streamlined case in their Introduction if they took a more open-ended view of key innovations from the onset and discussed the evolutionary models (early burst and pulsed) that are consistent with scenarios on how different kinds of key innovations might operate. Doing so changes the focus from whether venom is a key innovation, which ultimately depends on one's definition of the term, to how venom has evolved and influenced the evolutionary radiation of venomous snakes.

Thank you for this important feedback. We agree with the reviewer and thus have replaced the section about using gene expression data (which had its own shortcomings), with a section talking more about the different evolutionary models and their relevance in studies of key innovations, their limitations, and the use of pulsed models to account for the limitations.

We include the following:

Ln79-98: Evolutionary models are extensively used to study trait evolution, and have been used to model everything from body shape evolution to gene expression level evolution (Freckleton et al. 2003; Brawand et al. 2011). Therefore, it is not surprising that evolutionary models are also widely used to study key innovation. However, rarely does one model consistently explain the evolution of key innovations (or traits believed to be key innovations). Some traits are better explained by brownian motion (BM), others by Ornstein-Uhlenbeck (OU) models, some traits fit a single-peak OU model better, while others a multi-peak OU model, other traits fit neither BM or OU processes well (Price et al. 2010; Colombo et al. 2015; Poe et al. 2018; Burress et al. 2019; Arbour et al. 2019). Along with BM and OU models, it is also possible to model early burst (EB). An EB in speciation rate and trait evolution is believed to be the predominant pattern in adaptive radiation (11–13). While it is possible to model early burst evolution, evidence for it is rarely observed in comparative data (14). The often conflicting results between these models require cautious interpretation of features like evolutionary rates (Price et al. 2010). Perhaps one limitation of these models is using a Gaussian process to model continuous trait evolution. Evolutionary processes can result in changes that are too abrupt to be accounted for by a Gaussian process (Landis et al. 2013). Pulsed models offer one solution. Utilizing a Levy process, the pulsed models can account for abrupt shifts in the continuous character evolution, that conventional evolutionary models cannot easily explain (Landis et al. 2013). Using this approach Landis and Schraiber found that body size evolution is better represented by rare stochastic pulses of diversification than convention EB or multi-optima OU models (Landis and Schraiber 2017; Martin and Richards 2019). Therefore, examining traits using a pulsed model of evolution might reveal previously unresolved trends of evolution.

Specific Comments

Title

The title summarizes the main results of this study about the tempo and mode of toxin evolution in snakes, but it does not fully convey what the paper is really about, which is whether snake venom is a key innovation and how it has influenced the radiation of snakes. Perhaps the title ought to reflect the full scope of the paper.

We appreciate the reviewer's encouragement here. However, we decided to take a conservative stance and have the title reflect only our main results. Since we're trying to challenge a somewhat

strong belief in the field, introducing the challenge with a narrative build rather than with a strong claiming title would allow a gentle transition to the reader and not impose a bias from the off-set.

Abstract

Line 40. "are essential for prey capture" should perhaps be "facilitate prey capture". The latter wording is more active. Also, venom is essential only if a snake is restricted to a certain predation strategy (strike, follow, and wait) that requires it. Other snakes (boas and pythons) can also capture prey without venom, but with a different strategy and adaptations.

Agreed. We have modified the abstract.

Line 51. Change "influence" to "influences". Insert "in" after "process" and "that" after "way". Also, "magic traits" is an odd phrase that requires explanation or at least a reference. Therefore, it may be better used in the main text than here in the abstract where it can neither be explained nor a reference cited.

We have modified this line and removed 'magic traits'.

Introduction

Line 60. "ecological speciation". Some authors (e.g., Mike Rosenzweig) have used this term to mean a kind of sympatric speciation, whereas others do not and treat it as a kind of allopatric speciation. The authors should probably explain what they themselves mean by using the term.

Done.

Ln72-73: Ecological speciation, i.e. speciation driven by differences in ecology, is considered the primary mode by which adaptive radiation can take place, and as various traits produce specific differences in ecology, certain traits are more strongly associated with the radiation process than others (3,11)

Lines 67-70. "Rather, the role of key innovations should be restricted to providing entry into novel ecological niches or adaptive zones, and studies should aim to identify specific shifts in tempo and mode of phenotypic evolution of the assumed key innovation (9,10)." If the authors wish to adopt this restricted definition of key innovation, or more precisely criteria for recognizing a key innovation, that is fine. However, if adopting this definition leads them to use the Early Burst model as the sine qua non of a key innovation, as they seem to do in the following paragraph, then they undermine their own argument. Ultimately, the authors find that the evolution of venom toxins in snakes is better explained by Pulsed models than by an Early Burst model. That does not surprise me because venom toxin seems like a evolutionarily versatile system as the authors describe it, a lot like the pharyngeal jaws of cichlid fishes in sense of facilitating further, rapid evolutionary change. Finding pulsed rate of evolution is what I would expect of a key innovation that promotes further evolutionary change.

We thank the reviewer for this feedback. As mentioned above, we have streamlined our argument more by focussing on evolution of the snake venom phenotype itself rather than by estimating shifts in evolutionary rates and testing various trait evolution models.

The next thing that I would want to know is what are the tradeoffs. Are there any costs to having venom? How about certain combinations of toxins? I would advise the authors to frame their argument about venom as a key innovation in comparison to other innovations that are versatile and highly evolvable. Such adaptations tend to be those with low costs to the organisms, often through relaxed trade-offs among functions.

This is an important point that we actually decided to include in the conclusion. Doing so reinstated our motivation of using snake venom to study the evolution of key innovations.

Ln382-389: Snakes usually need to produce large amounts of venom, and determining if venom is more costly compared to other offensive (or defensive strategies) is difficult, as it requires prey-handling experiments, taxon specific toxicity testing, etc which are both complicated, and difficult to implement (Jackson et al. 2019). However, considering the several ways snakes can modulate venom output (eg venom metering, secretions with reduced protein content etc), venom might actually be an effective way of procuring energy rich meals (by subduing large prey) making it a particularly cost-effective innovation (Schendel et al. 2019; Jackson et al. 2019).

Line 85. "rarely observed in data (14)." The authors should clarify what kind of data. Comparative data in extant clades, I should think. In the fossil record, the early burst pattern is more commonplace but by no means universal.

Ln88-89: Have modified the text.

Lines 85-99 "This lack of empirical support might be because studies are overlooking the components of traits that actually produce phenotypic change; gene expression variation" and continuing to the end of the paragraph. I did not find this particular argument to be a convincing explanation for why few examples of early burst adaptive radiations occur among extant species. I do understand the authors' main point that rates of evolution of genes and phenotypes may differ, and I understand the authors' need to motivate their study around a specific question and a hypothesis. However, in adaptive evolution, genes are only going to evolve as rapidly as the phenotypes to which they correspond are sorted by natural selection. One way in which genes might evolve faster is in drift among selectively neutral variants. But probably would not be the case here. Alternatively, and perhaps more of a problem, is in cases of highly developmentally canalized systems in which the same phenotype can develop from more than one, and perhaps multiple different gene expression patterns. This situation may be commonplace and could lead to higher rates of evolution in genes than in phenotype.

We agree with the reviewer here and indeed feel our original motivation might have been a bit forced. As the reviewer showed interest in the study of the venom trait itself, we have decided to shift focus exclusively to understanding the evolutionary dynamics of gene expression in snake venom toxins. We have modified the introduction as mentioned previously (Ln79-98).

Lines 100-127. The problem of gene expression-phenotype mapping in complex traits versus in snake venom. The authors do make a good case here for the special status of venom because of its simplicity, in which different levels of gene expression result directly in different proportions among toxins in the venom “cocktail”. In other words, in snake venom one need not consider gene networks, morphogenetic pathways, and the like. The system sounds ideal. However, several thoughts occurred to me as I read this passage. What are the observed performance differences among toxin cocktails? What specific associations exist between toxin mixtures and either prey or environments? The idea that toxins do not interact with other is presented rather definitively and with citation of just one supporting reference. How was this inference made?

Thank you for the comment. In an attempt to answer the reviewer’s comments we have included the following in the introduction:

Ln116-122: Venom toxins can have both agonistic and antagonistic interactions between other toxin components, but how they influence other traits outside the venom system is unclear. On one hand venoms are integrated systems with different toxins acting in concert to immobilise prey (Reeks et al. 2015). On the other hand, whether this mode of action introduces an evolutionary constraint is less clear, since there is little phylogenetic covariance between components, and gene-environment constraints appear to act on individual loci, independent of co-expression patterns between toxin genes (Barua and Mikheyev 2019; Zancolli et al. 2019).

Although the authors present toxins within venom as isolated adaptations without interactions, venom requires a delivery system (glands and fangs) as they later mention briefly, a specific hunting strategy (behavior and supporting morphology, including sensations), and either immunity from the effects of one’s own toxins, some barrier sequestering toxins, or at least storage of toxins in an inactive state until deployed. How is all this potential complexity managed? One could imagine far-reaching consequences of having venom or of variation among venom cocktails on the phenotypes of snakes. Is venom really such an isolated system?

The reviewer is right in saying that the venom system has individual components which require a certain degree of coordination to function effectively. We didn’t mean to undermine these components, but rather stress on the fact that there is no quantitative evidence of how these components interact. Whatever associations exist, are for the most part, speculation. But as the reviewer mentions, there could well be numerous interactions which could be tested. Perhaps in a future publication. For the current study, we include the following which acknowledges the above mentioned interactions.

Ln350-354: The venom system comprises venom toxins, venom glands, fangs, and muscle architecture responsible for delivering the venom into the prey. Although the toxins are directly involved in prey immobilization, evolution of the other non-toxin components are essential for development of the venom system as a whole. Numerous examples exist of toxin recruitments coinciding with development of various morphological features like high-pressure venom delivery, and certain hunting strategies like ambush feeding (Fry et al. 2008). However, any modifications to enhance prey procurement would be restricted to the venom system, and unlikely to affect changes in other parts of the animal (33).

Lines 133-135. "If snake venoms are key innovations we would expect to see shifts in their evolutionary rates in response to changes in optima." How are optima identified? Ideally, this would be by reference to evidence external to the snakes themselves, such as the timing of environmental changes from the paleoclimatic record, biogeographic dispersal events into new regions, or time of contact with novel prey. There is a certain circularity to inferring the existence of optima from the evolutionary rate dynamics of snake venom alone. Is there external evidence too?

Thank you for this feedback. We agree that the mention of optima in the introduction does seem a bit abrupt and out of place. For that reason we have removed it from the introduction, and have dedicated a paragraph in the discussion to focus on the type of optima and relevant evidence for the same:

Ln284-291: The high variability in the snake venom phenotype is likely due to the presence of various optima. A previous study showed that distribution of toxin families on the macroevolutionary scale can be explained by the presence of convergent phylogenetic optima (27). Furthermore, the effect of various environmental factors like temperature and longitudinal climatic gradient influence venom variation, hinting at the occurrence of optima that maintain disparate, locally adaptive venom complexes (28). Therefore, the shifts in phenotypic macroevolutionary rates are likely due to shifts between these optima.

Results

Lines 232-235. "Jump models have highest weighted AIC scores and are a better fit to snake b venom gene expression data as compared to conventional incremental models of evolution (Table 1; electronic supplementary material)." Are the jump models grouped together in Table 1 under the heading "Pulsed"? If yes, then please state so.

We have modified the table legend to reflect this. Thank you.

In general, yes, the "Pulsed" models all fit better than the random (BM and OU) and early burst (EB) models. Do the pulsed models fit significantly better? Also, can the "pulsed" models

distinguish oscillation between stasis and random walk, on the one hand, from sporadic pulses of directional change, on the other hand? In order to relate these models to those traditionally used to study evolutionary rates in paleontology (stasis, random walk, and directional evolution), it would be beneficial if the authors would explain the correspondences between these models and traditional ones.

In the methods we describe how the pulvR models different types of pulse processes: The Lévy process is represented mathematically using the Lévy-Khinchine representation, where one can compute variance of trait change along a branch of length t . We model stasis followed by rapid adaptation using a compound Poisson process. This is the Jump Normal (JN) process, which assumes jump sizes are drawn from a normal distribution. The other pulsed evolution model is Normal Inverse Gaussian (NIG) which uses an infinitely active Lévy process to model constant rapid adaptation.

We believe the JN is analogous to periods of stasis and random walk, while NIG is similar to sporadic pulses, but much more rapid. We are unsure whether there is any directionality per se. The pulsed models test for dynamics which are very different from conventional models. For example, it can test the effect of fluctuating population size on quantitative trait evolution, which would be impossible in conventional models (Landis et al. 2013). Furthermore, the package estimates BM and OU processes in the standard framework (by rescaling branch lengths as a function of model parameters) and then allows models comparison with pulsed models, to determine which model best suits the given data. In both (Landis and Schraiber 2017) and (Landis et al. 2013), the authors use simulation to check the effect of model biases, and whether parameter estimates differ significantly from that of a BM process. In both instances the authors found that the model parameters are consistently significant and have low model bias.

The intricacies and nuances of the Levy process and how it models evolution are indeed fascinating, but beyond the scope of our discussion. We mentioned in the supplementary (which has all the models estimates), that readers interested can refer to the appendix of (Landis and Schraiber 2017), which is much more detailed.

Lines 273-281. Optima, prey diversity, and environment. So far, the existence of optima has only been inferred in the authors' data from the good fit of the data to pulsed (Lévy process) models that assume optima. The authors could make a stronger argument if they could be more specific about what the optima are and how venom has evolved specifically to allow snakes to adapt to these shifting optima.

We have clarified this in the discussion (Ln284-291).

If specific associations are lacking, then at the very least they could explain how prey diversity and environment could in theory co-define optima as the authors claim. For example, what toxin combinations are better suited to a diverse range of prey, and which are better suited to one or a few prey types? Do venomous snakes conform to the specialist/generalist dichotomy? Are there

tradeoffs between toxin efficacy on one kind of prey and the number of prey types on which it can be used? How does the environment factor in? Through general levels of metabolic and behavioral activity expected of an ectotherm? Or in more specific ways through prey availability, overall or seasonally? Right now, these optima are just a little too vague to be totally satisfying.

The points the reviewer mentions are important and definitely worth exploring. However, they are beyond the scope of the current paper which deals more with phenotypic rate dynamics, and not the effect of phenotypic variation in venom. The manuscript does include a section with reference to other studies that deal with this explicitly:

Ln124-127: Changes in expression levels of individual toxins alter their abundance in the venom, thereby influencing venom efficacy (39–41). This alteration in venom efficacy impacts the feeding ecology of snakes, which in turn determines how snakes adapt and colonize new niches (42,43).

As we have explained the origin of the optima in previous sections, we hope this satisfies the reviewer's queries.

Lines 346-349. "Nearly all studies of adaptive [change to "adaptation"] focus on traits and processes in extant species, [and] this is a major disadvantage since there is no way of representing extinct taxa and thus no way of determining whether a clade with specific innovations was more species[-]rich in the past (77,78)." Well said, and very refreshing to see this statement here. Please note minor edits in brackets.

Thank you for the comment. We have made the appropriate edits. (Ln370-371).

We thank the reviewer for their interest in our work. We hope the changes made are satisfactory, and welcome any new suggestion they might have.

Appendix C

Reviewer(s)' Comments to Author:

Referee: 1

Comments to the Author(s)

Firstly, I should disclose that I have already reviewed this manuscript elsewhere on a prior submission (to my knowledge this is no longer being considered there so there is no conflict), and I am glad to see that substantial changes have been made which, on the whole I think have improved the manuscript. Nevertheless, I still have outstanding concerns/comments here, partly issues that remain from my earlier review and partly new comments introduced in the current version. I think this manuscript is well placed to make an impact in the field if these comments are addressed, and so I list them below (in roughly chronological order) in the hope that they are useful to the authors.

Ln 41-42 - I think you need to clarify your comment that there have been 'no quantitative analyses' testing whether venom is a key innovation. There have been tests of the hypothesis that venom is associated with higher diversification rates, which is often regarded as a major component of key innovations and, more recently, has been regarded as a non-essential but (nevertheless) a common outcome of key innovations (and by many definitions is the 'radiation' part of adaptive radiation).

Thank you for the clarification. We modified the abstract to focus on rate dynamics instead:

Ln 40-42 - Venoms evolve rapidly, are essential for prey capture, and are widely believed to be key innovations leading to adaptive radiation. However, few studies have estimated their evolutionary rate dynamics.

Ln 51 - I'm not aware of any cases where someone has proposed that venom acts as a 'magic trait', merely as a key innovation.

Removed 'magic traits' replaced with:

In 51-52 - Therefore, the extent to which venom directly influences the diversification process is still a matter of contention.

Ln 108-111 and 119-120 - I'm not convinced by the relevance of your assertion that "highly tissue specific genes would also likely reduce significant pleiotropic constraints and cross-phenotype associations helping to discern the unique trajectories of individual genes". While tissue-specific expression would certainly reduce such interactions across tissues, many venom toxins are widely suspected or known to have lots of pleiotropic functional effects and synergistic or antagonistic interactions with other venom components. In other words, as a highly specialised and multidimensional (complexity of different interacting toxins) system I

don't think that venom molecular evolution can be expected to typically have low rates of pleiotropy or molecular interactions.

We agree with this point and have modified the text accordingly:

Ln 116-122 - Venom toxins can have both agonistic and antagonistic interactions between other toxin components, but how they influence other traits outside the venom system is unclear. On one hand venoms are integrated systems with different toxins acting in concert to immobilise prey (Fry et al. 2006). On the other hand, whether this mode of action introduces an evolutionary constraint is less clear, since there is little phylogenetic covariance between components (Barua and Mikheyev 2019).

Ln 128-129 - Depending on your definition of key innovation you could perhaps argue that phenotypic differentiation driven by ecology has not been studied well in snakes (although you could easily consider some of the work on diet and venom evolution to fall into this bracket), however, when talking about "the extensive radiation of snakes", which strongly implied lineage diversification, it is untrue to claim that "no studies have tested this hypothesis". In fact, I find it surprising that none of the work linking venom evolution to lineage diversification rates appears in this paper given it's clear relevance at various points. In the interest of full disclosure, some of this work is my own (e.g. Harris and Arbuckle, 2016, Toxins 8:193) but other work is not (e.g. Liu et al., 2018, Mol Phylogenet Evol 125:138-146).

Thank you for the suggestion. We have modified the text to include the following:

Ln 130-151 - The idea that venom is a key innovation and that it underlies the extensive radiation of snakes is pervasive in the literature (2,43–50). Yet, few studies have examined long-term changes in evolutionary rates of venom gene expression in snakes. There are numerous studies that have examined the role of venom in lineage diversification in other taxa (Harris and Arbuckle 2016; Arbuckle and Speed 2015; Liu et al. 2018; Blanchard and Moreau 2017). In blenny fish the presence of a venom system in the form of a buccal gland and fang is associated with higher rates of diversification (Liu et al. 2018). In tetrapods, evolution of venoms and poisons are typically associated with an increase in diversification rates (except in amphibians) (Harris and Arbuckle 2016). There is also a substantial amount of literature suggesting the role of diet in lineage diversification (Rainford and Mayhew 2015; Burin et al. 2016; Rojas et al. 2018). Since snakes use venom primarily for prey procurement, alterations in venom and diet could have an effect on diversification in venomous snakes.

Key innovations however have more features than just causing lineage diversification. Key innovations contribute to expansion of ecological ranges, they represent optimal adaptations, and usually undergo changes in evolutionary rates to fill morphospace (Givnish 2015). Restricting the role of key innovations to just diversification ignores these features, and removes emphasis on evolution of the key innovation itself (Givnish 2015; Rabosky 2017). In this study, we specifically focus on the evolution of snake venom. We use a comparative dataset of snake venom gene

expression to identify shifts in macroevolutionary rates, which are characteristic of key innovations (Rabosky 2017). To further characterize patterns of venom evolution, we estimated long-term changes in evolutionary rates of venom gene expression, and also fit the data to several trait-evolution models. Our results revealed that toxin expression in snake venom evolves very rapidly, and has experienced numerous shifts in evolutionary rates over the past 60 million years.

Ln 135 - optima for what?

We have removed this line as it is confusing.

Ln 145-147 - How did you deal with cases of multiple studies of the transcriptomes for a single species?

For consistency, we use the previously collected and carefully curated dataset from (Barua and Mikheyev 2019). Since this data was collected in 2017, it doesn't include studies published after that.

Ln 159 - Fig 1 seems to be missing. I am unclear from the wording here whether you are referring to Fig 1 within the Supplementary Material or both Fig 1 and Supplementary Material, but in either case there is no Fig 1 in your manuscript (numbering starts at Fig 2).

Should be in the electronic supplementary material. We will double-check next time if it was uploaded properly.

Ln 163-165 - What is the relevance to your study of age of the clade in your study? It seems an odd detail to add with no explanation of its relevance.

Since we use a time calibrated tree we decided to mention the estimate of the root, which some readers find useful as it gives them a perspective of time. We have not made any modifications to the text in regard to this.

Ln 180 - "toxins families" should be 'toxin families'

Ln -194 - Have fixed in the manuscript.

Ln 183-192 - There are a few points of interpretation of your data that I'm not sure you have considered (at most having a couple of partially related comments later in the manuscript for some of these points but not fully incorporated or explored). Of the toxins with the best evidence of 'early burst' patterns (which are potentially indicative of adaptive radiation) two important toxins stand out - SVMs and TFTs. Specifically, these are often the most abundant toxins, are often toxicologically dominant ones, and the 'early burst' patterns are largely present in vipers (for SVMs, colubrids also show this pattern) and elapids (for TFTs, also stated on Ln 211), which also corresponds to the main toxin classes in each of these lineages. The

alternative lineages for these toxin patterns may be evidence for trade-offs (as shown previously by the authors - in practice if not necessarily these toxin classes appear as 'alternative' venom composition types).

Thank you. Following this suggestion we decided to rethink the rate through time analysis. We decided not to use rate through time plots to infer early burst, considering that no study has used it in that manner, and as we now feel it isn't quite capable of characterising early burst evolution well. Instead, we focus specifically on observing rate patterns in venom expression evolution in the three families. We test for early burst evolution using the modelling approaches discussed in the next sections, which provide a more direct comparison of various alternative hypotheses. We have modified text accordingly:

Ln 208-212 - Under adaptive radiation hypothesis, ecomorphological rates should transition from rapid rates to slow, equilibrium rates as ecological niche gets filled (Slater et al. 2010). To identify these patterns we estimated the rates of toxin expression evolution of each toxin family after the split of the three families. Our estimates of 'rate through time' revealed that toxin families show unique evolutionary rates and rate dynamics in each snake family.

Ln 228-231 - SVMP, TFTx show an interesting pattern where they seem to represent alternate venom types. SVMP has high rates, and is dominant in vipers (and to an extent in colubrids) while TFTx has higher rates, and is predominant in elapids. The alternate lineage of these toxins could be evidence of tradeoffs, a pattern that we previously observed (Barua and Mikheyev 2019).

Importantly, the time-scale of toxin origins in relation to your clade under study is important for interpreting patterns of evolutionary rate dynamics. In your results, some toxins start off with very low rates and show patterns consistent with early burst style dynamics (e.g. C-type Lectins and SPs) or maintain low rates across most of the tree (e.g. KSPIs). As these are predicted to pre-date the root of your tree they may have already experienced an early burst before slowing prior to the earliest point of your study. Consistent with this idea is that the end of the early burst style patterns of SVMP and 3FTxs have estimated rates comparable to the root estimates of other toxins. For some of the patterns you are looking for it is important to have the origin of the trait within the phylogeny and this important caveat is neither stated nor discussed (the closest is on Ln 296-298 but this failed to acknowledge that almost all the toxin families here pre-date your clade, and most of them originated on one of two branches).

Thank you for raising this important point. You are absolutely correct that ideally the traits should have originated within the time points of our tree, and that it is perfectly plausible that these toxins may have experienced the early burst pattern before the earliest point in our tree. However, we are more interested in their recent role in evolution. All of the lineage divergence in our tree occurred well after the root. Therefore, if the traits were already present before the diversification, they might not have contributed substantially to the diversification of extant lineages (which is the predominant belief in the field). Rather than recruitment, if changes in

expression level of the trait contributed to diversification, we would expect to see the patterns that we observe in toxins like C-type Lectins and SPs.

We have expanded the discussion to to incorporate this likely caveat:

Ln 312-323 - Rate dynamics and age also tend to be related, with older toxins experiencing higher rates in the past followed by reduction in modern lineages (Fig 2). It should be noted that the probable origin of most of the toxin families in our study pre-dates the root of our tree(Fry et al. 2006). Typically, if a trait is responsible for lineage diversification, its origin should be at a point within the tree around the time of major branching events. However, we can only examine what happens to venom evolution after the most recent common ancestor of extant snakes. Some of the oldest toxins to comprise the venom: SVMP, TFTx, and CRISP (Fry et al. 2008) all showed a larger distribution of high evolutionary rates near the root than the tips (Fig 2). This could be because the major toxin families were likely present in the ancestral venom and experienced a uniform reduction in evolutionary rates as lineages diversified. These toxins that pre-date the root likely allowed ancestral snake lineages to realize their ecological potential which lead to niche specification, which in turn lead to a slowdown in evolutionary rates in their descendants.

Ln 195-197 - How are you defining "major lineages" in the context of 'expectations that rates of trait evolution should be high immediately after they diverge'? Why is the split of the three venomous families an important evolutionary point for your study (note also that you should check if 'Colubridae' is actually the family you mean as this 'traditional wastebasket taxon' has been split into many families)? What's special about the divergence of these particular clades compared to any other clades on the tree?

For this we actually use a convention laid out in the BAMM tutorial under the 'rate through time' section (<http://bamm-project.org/postprocess.html#rate-through-time-analysis>). This way of computing rates from a specific point of interest in the tree is common and widely used(Rabosky et al. 2014; Fernández-Mazuecos et al. 2019; Folk et al. 2019). We selected these nodes specifically because they are the points of origin of the three major venomous snake families in our tree.

We agree that classifications of 'Colubridae' are sometimes unclear. For our study we broadly classify the non viper and non-elapid lineage as colubrids. Since we used the same tree and same classification in our previous study, we decided to do the same for consistency.

Ln 201 - Colubrids shouldn't be capitalised

Ln 214 - Have fixed the error.

Ln 206-207 - This time period is approximately coincident with the colonisation of the New World by vipers, which is potentially consistent with venom evolution being linked to ecological opportunity.

We modified the text to include this:

Ln 218-223 - In vipers, toxin families generally showed an increase in evolutionary rates, with a majority of them occurring at around the 20-million-year mark, just after the diversification of the major viperid lineages, which is potentially consistent with venom evolution being linked to ecological opportunity.

Ln 207-208 - You claim that "SVMP...showed an increase in evolutionary rates since the divergence of vipers", but Fig 3 seems to contradict this.

Correct. It's my mistake. Have deleted the sentence.

Ln 210 - "Ohiophagus hannah" should be 'Ophiophagus hannah'

Ln 225 - Done.

Ln 219 - There should be an 'and', not a comma between "Brownian motion (BM)" and "Ornstein-Uhlenbeck (OU)".

Ln 237-238 - Yes. Corrected.

Ln 220 - Why isn't Early burst a 'conventional' model? It's been a fairly standard part of the set of phenotypic evolution models for a pretty long time now.

Ln 237-238 - Agreed. We have clubbed them together.

Ln 292-294 - This again suggests that the most abundant (and often toxicologically dominant) toxins are likely the most important links to the ecology and evolution of the animals and their venom systems - they are probably more exposed to selection as a result.

Agreed. We have modified the text to include this suggestion:

Ln 307-311- Abundant toxin families along with ePLA2 and SVSP showed a higher net rate of evolution of gene expression (Fig. 3), a trend also observed in sequence data (Aird et al. 2015, 2017). This suggests that the most abundant (and often toxicologically dominant) toxins have the strongest links to the ecology and evolution of snakes and their venom systems - they are also probably more exposed to selection as a result.

Ln 298-299 - You didn't estimate ancestral character states (or at least don't present them in the manuscript), you estimate rate dynamics.

Correct, we have removed the sentence.

Ln 306-307 - Why not test for trait-dependent diversification rather than simply say 'it is tempting to speculate on this but there is a lack of evidence' (and, as above, you might want to discuss what evidence does exist for venom/toxin-associated diversification). You again say something similar on Ln 314-315 in that venom 'might not necessarily...show patterns of trait-dependent diversification', well, not necessarily but you could test for it.

Testing trait dependent diversification for continuous traits required a large amount of data which is frankly not available in our case. Also, along the lines of our narrative, we do not want to test for trait dependent diversification because we believe it's a restricted perspective of key innovations (as mentioned before). The predominant belief that we are trying to challenge in that key innovations provide ecological opportunity, but not necessarily lead to lineage diversification. When looking at patterns of trait evolution many of the dynamics are attributed to causing species divergence. We want the reader to appreciate that the patterns we observe are characteristic of key innovations, but these should not be considered as evidence for trait dependent diversification. We want to stress on the fact that species divergence does not happen through an isolated mechanism, but requires many specific conditions, which may be difficult to meet (as mentioned in the discussion). Therefore, while it is easy to interpret our observations as evidence of species diversification, one should not jump the gun without considering all the possible lines of evidence. A similar argument has been made in this recent study, with a much larger sample size, from which we can draw parallels(Phuong et al. 2019).

Ln 312-313 - phenotypes don't lead to key innovations, phenotypes are the key innovations (the evolutionary consequences of which are what makes the difference to terminology)

Ln330-331 - Apologies for the misconception. Have clarified in the text.

Ln 329 - What do you mean by 'large-scale' diverse forms?

We mean that evolution of modular traits would likely not be associated with or cause evolution of other traits. For example, venom evolution would not lead to morphological changes outside the venom system, like developing appendages.

Ln 338 - It is very unlikely that snakes show mate choice based on venom composition, but that very direct mechanism is not necessary for venom phenotypes to "cause widespread character changes that are needed to establish reproductive isolation that leads to speciation". Indirect consequences would still ultimately be due to venom variation, for instance ecological separation (complete or partial) relating to diet specialism, foraging style, related habitat choice (for particular prey), differences in activity (better defended species may be less constrained by predation, be more active, and hence achieve a higher frequency of mating opportunities). As you allude to, little direct evidence for any of these types of mechanisms exist, but the important part is that neither direct venom-based mate choice nor widespread character changes (it could be one or two simple but important changes) are necessary.

Thank you for this insight. We have included this in our text as it makes our argument more complete.

Ln358-363 - It is not known if snakes have a similar kind of mating preference based on venom composition. For example, would a female prefer a male with more similar or dissimilar venom composition for mating? Venom could however, lead to indirect ecological consequences in terms of foraging style, habitat choice, temporal differences in activity. But speciation requires a level of reproductive isolation, how this is achieved either directly or indirectly by changes to the venom is not obvious.

Ln 346 - I think you mean 'adaptation' or 'adaptive radiation' instead of just "adaptive".

Ln369 - Thank you. We have fixed it.

Ln 347-348 - Although you are correct that there is no way of "determining" (with absolute certainty) whether a clade with particular traits was more diverse in the past, there are plenty of ways to estimate this, so it's not quite so much of a lost cause as you suggest.

Sure, we'll try and tone down the cynicism there :-)

Ln 351-352 - Can you suggest plausible examples of such different conditions and processes that are relevant to venom evolution? Unless we have evidence of different processes we typically assume some degree of uniformitarianism (without which any historical science is impossible).

We added this line as a general caveat to these kinds of studies. We included the following in the text:

Ln373-380 - While most studies provide a microevolutionary perspective, extrapolating from processes that operate in the present day to what happened early in a clade's history is difficult; because conditions were different in the past, different processes may have been at work, or may have produced different outcomes. Perhaps in the past there were venomous snakes with venom compositions specific to the past environment. In response to any changes in this environment snakes could have evolved venom composition starkly different from the ones we see today. There might also have been venomous snake lineages in the past that became extinct leaving a whole history of venom composition unexplored.

Ln 354 - Should "analysis" by 'analyses'?

See next comment.

Ln 355-356 - Just because you are necessarily basing your interpretations on the data you collected doesn't mean that your interpretations and findings are specific to your dataset

only...at least I hope not as this would undermine the biological insights your study could give (or indeed any study ever published).

Thank you for this input. We agree and have removed the sentence. We realize that it didn't read well, and didn't convey our intended message.

Ln 360-362 - I would have thought that the locations of rate shifts are far more relevant to your hypotheses than the number of rate shifts? Would your hypotheses be better supported if you found 6 vs 3 vs 1 shifts at random points in the phylogeny or particular rate shifts that occur at locations consistent with related evolutionary events you think are important?

While the location of the rate shifts are important, the way BAMM works, it's advised not to stress specifically on the location of the rate shifts:

"In the BAMM framework, there is no single set of independent rate shifts waiting to be identified. Rather, BAMM identifies configurations of rate shifts - sets of shifts that are sampled together -

The model assumes that a relatively small number of discrete shift events can explain the data (sort of.... it is more that the marginal likelihood of any particular model is implicitly penalized by the addition of more parameters).It is possible that many shifts in evolutionary dynamics change in a discrete fashion (e.g., the classic "key innovation" scenario), but it is also possible that major changes in dynamics occur through a number of sequential changes in some general region of a tree"

<http://bamm-project.org/rateshifts.html>

However, if rate shifts occur in unexpected, spurious locations (like they do for ePLA2 and TFTx in Crotalus), they have to be explained.

Ln 371 - "misrepresented" should be 'misrepresented'

Fixed.

We have also added the caveat sections to the supplementary as it allows a greater exploration of caveats and their solutions without being limited by the word limit of the journal. We have mentioned this in the relevant methods section of the manuscript.

Ln 374 - you have doubled up the words "by the"

Fixed.

Ln 384-388 - I don't follow why a procedure applied to all species would only affect this one particular case.

This is because these two Crotalus species are very closely related and have very different venoms from one each other (one has more structural zeros than the other). The normalization (scaled around the mean) is done within species and this inflates the differences between them. Like mentioned this is appropriate for toxins that are actually present in both the snakes. But if a toxin is absent in both the snakes, its normalized value would depend on the value of the other toxins that are present. This creates an artificial difference between both the species for the missing toxin.

If the normalization was causing spurious differences throughout the dataset we would see it in other closely related species as well; Protobothrops, which like Crotalus do not have ePLA2, or Naja, and Micrurus, which do not have vPLA2. Since we do not see spurious shifts in these lineages, we can be fairly certain that the normalization only had an effect in the two Crotalus species.

Ln 403-405 - I agree that considering "the evolution of snake venom in terms of its impacts on speciation" would provide insight into venom as an influence in adaptive radiation, but as mentioned earlier, this has been tested in some other venomous groups (and also including, in combination with other tetrapods, in snakes). The existing studies focus on the presence of venom, rather than specific attributes, so there is more to be done here, but I think the lack of any discussion of this literature is a bit of an oversight (given you refer to the basic questions it aims to address at multiple points in your manuscript).

Agreed. As mentioned before, we have included references to several papers that talk about this.

Ln 421-424 - How did you combine these two trees, and why did you choose to use two different phylogenies?

We used an already published time-calibrated phylogeny from (Zheng and Wiens 2016). This phylogeny is one of the best ones in literature and is perfect for large comparative analysis.

Ln 428-429 - something has gone a bit wrong with superscripts here.

Ln431-432 - Thank you. It is fixed.

Ln 430 - "conservatiove" should be 'conservative'

Ln433 - Thank you. It is fixed.

Ln 433 - By "significant from" do you mean 'significantly different from'?

Ln 436 - Yes, thank you.

Ln 434 - Convergence doesn't need to be capitalised.

Ln437 - Thank you. It is fixed..

Ln 443 - What do you mean by an "explosive EB" model? How is this different from any other EB model?

Ln449 - It is not different,we have fixed this.

Ln 449-450 - I'm not sure what the sentence beginning with "one of the models" is saying, the wording is very confusing.

Apologies. We have modified the sentence:

Ln 455-457 - We model stasis followed by rapid adaptation using a compound Poisson process. This is the Jump Normal (JN) process, which assumes jump sizes are drawn from a normal distribution.

Ln 459 - When you say "weighted AIC was used as a measure of model fit" I'm not sure what you mean - do you mean AICc (which isn't mentioned elsewhere) or Akaike weights (aka model probabilities, which looks more similar to what is presented in ESM Table 1, but in this case I don't understand why the values in that Table don't sum to 1 in each row)? Particularly if you mean Akaike weights, these aren't a "measure of model fit" as such, but merely the probabilities of each model being the best in your model set (they say nothing about how good your model set is - none of them could fit well but you'll still get some better than others).

Thank you for the clarification. Yes indeed, AIC weights do not represent model fit, but they are used to select the best performing model(Wagenmakers and Farrell 2004). We use the AIC weight to determine which model best suits our data. The values in our table represent AIC weights for each of the 9 models we tested (BM, OU, EB, and 6 pulsed models). In all cases the pulsed models were favoured as compared to the non pulsed models. However, each pulsed model had very similar weights, which make it difficult to determine which pulsed model is better. For that reason we club them together and report the highest AIC weight.

Ln 460-461 - The absolute values of AIC are meaningless for interpretation, and hence their ratio is as well, so I assume it's just unclear what you mean here but a model with twice the AIC of another model says nothing about the relative evidence each provides.

Ratio of AIC weights are used to determine the best model among a group of models(Wagenmakers and Farrell 2004). Our use of a criterion for a better model follows the approach used in (Landis and Schraiber 2017).

.....

I hope the authors aren't too discouraged by these comments, and (as I said earlier) the manuscript is certainly in better shape now than the earlier version I reviewed. I do believe the basic idea and analyses here have something important to offer the field, but my comments are intended to help improve the presentation and interpretation of these results. I hope the authors find them useful.

We appreciate the reviewer's continued investment into the quality of this study.

Best wishes,
Kevin Arbuckle

Refs

- Aird, Steven D., Shikha Aggarwal, Alejandro Villar-Briones, Mandy Man-Ying Tin, Kouki Terada, and Alexander S. Mikheyev. 2015. "Snake Venoms Are Integrated Systems, but Abundant Venom Proteins Evolve More Rapidly." *BMC Genomics* 16 (August): 647.
- Aird, Steven D., Jigyasa Arora, Agneesh Barua, Lijun Qiu, Kouki Terada, and Alexander S. Mikheyev. 2017. "Population Genomic Analysis of a Pitviper Reveals Microevolutionary Forces Underlying Venom Chemistry." *Genome Biology and Evolution* 9 (10): 2640–49.
- Fernández-Mazuecos, Mario, José Luis Blanco-Pastor, Ana Juan, Pau Carnicero, Alan Forrest, Marisa Alarcón, Pablo Vargas, and Beverley J. Glover. 2019. "Macroevolutionary Dynamics of Nectar Spurs, a Key Evolutionary Innovation." *The New Phytologist* 222 (2): 1123–38.
- Folk, Ryan A., Rebecca L. Stubbs, Mark E. Mort, Nico Cellinese, Julie M. Allen, Pamela S. Soltis, Douglas E. Soltis, and Robert P. Guralnick. 2019. "Rates of Niche and Phenotype Evolution Lag behind Diversification in a Temperate Radiation." *Proceedings of the National Academy of Sciences of the United States of America* 116 (22): 10874–82.
- Fry, Bryan G., Holger Scheib, Louise van der Weerd, Bruce Young, Judith McNaughtan, S. F. Ryan Ramjan, Nicolas Vidal, Robert E. Poelmann, and Janette A. Norman. 2008. "Evolution of an Arsenal: Structural and Functional Diversification of the Venom System in the Advanced Snakes (Caenophidia)." *Molecular & Cellular Proteomics: MCP* 7 (2): 215–46.
- Givnish, Thomas J. 2015. "Adaptive Radiation versus 'Radiation' and 'Explosive Diversification': Why Conceptual Distinctions Are Fundamental to Understanding Evolution." *The New Phytologist* 207 (2): 297–303.
- Harris, Richard J., and Kevin Arbuckle. 2016. "Tempo and Mode of the Evolution of Venom and Poison in Tetrapods." *Toxins* 8 (7). <https://doi.org/10.3390/toxins8070193>.
- Landis, Michael J., and Joshua G. Schraiber. 2017. "Pulsed Evolution Shaped Modern Vertebrate Body Sizes." *Proceedings of the National Academy of Sciences of the United States of America* 114 (50): 13224–29.
- Liu, Shang-Yin Vanson, Bruno Frédéricich, Sébastien Lavoué, Jonathan Chang, Mark V.

- Erdmann, Gusti Ngurah Mahardika, and Paul H. Barber. 2018. "Buccal Venom Gland Associates with Increased of Diversification Rate in the Fang Blenny Fish *Meiacanthus* (Blenniidae; Teleostei)." *Molecular Phylogenetics and Evolution* 125 (August): 138–46.
- Phuong, Mark A., Michael E. Alfaro, Gusti N. Mahardika, Ristiyanti M. Marwoto, Romanus Edy Prabowo, Thomas von Rintelen, Philipp W. H. Vogt, Jonathan R. Hendricks, and Nicolas Puillandre. 2019. "Lack of Signal for the Impact of Conotoxin Gene Diversity on Speciation Rates in Cone Snails." *Systematic Biology* 68 (5): 781–96.
- Rabosky, Daniel L. 2017. "Phylogenetic Tests for Evolutionary Innovation: The Problematic Link between Key Innovations and Exceptional Diversification." *Philosophical Transactions of the Royal Society of London. Series B, Biological Sciences* 372 (1735). <https://doi.org/10.1098/rstb.2016.0417>.
- Rabosky, Daniel L., Stephen C. Donnellan, Michael Grundler, and Irby J. Lovette. 2014. "Analysis and Visualization of Complex Macroevolutionary Dynamics: An Example from Australian Scincid Lizards." *Systematic Biology* 63 (4): 610–27.
- Slater, Graham J., Samantha A. Price, Francesco Santini, and Michael E. Alfaro. 2010. "Diversity versus Disparity and the Radiation of Modern Cetaceans." *Proceedings. Biological Sciences / The Royal Society* 277 (1697): 3097–3104.
- Wagenmakers, Eric-Jan, and Simon Farrell. 2004. "AIC Model Selection Using Akaike Weights." *Psychonomic Bulletin & Review* 11 (1): 192–96.
- Zheng, Yuchi, and John J. Wiens. 2016. "Combining Phylogenomic and Supermatrix Approaches, and a Time-Calibrated Phylogeny for Squamate Reptiles (lizards and Snakes) Based on 52 Genes and 4162 Species." *Molecular Phylogenetics and Evolution* 94 (Pt B): 537–47.

Appendix D

Referee: 2

Comments to the Author(s)

Review of RSPB-2020-0122

General Comments

This is an interesting and significant investigation of the macroevolutionary consequences of possessing venom in snakes. The study infers rates and patterns of evolution in toxin gene expression through transcriptome techniques. The data are mined from published work and assembled and analyzed here to test various evolutionary pattern and process models for goodness of fit. In the end, the authors demonstrate that pulsed models, in which rates of evolution vary through time as if in response to shifting optima, fit the data better than random and early burst models. The core of the study is bracketed between an Introduction and a Discussion and Conclusion interpreting venom as a key innovation within a framework of adaptive radiation.

Overall, the paper is well written, genuinely compelling, and shows an understanding of many kinds of data from multiple subfields of evolutionary biology. There is something in this paper for everyone. The authors demonstrate an awareness of and acknowledge the limitations of their own data. They also are aware of the strengths of other kinds of data.

The overall rationale for the study, as presented here, is to test whether gene expression data conform to an early burst model of evolution better than phenotypic data. The authors present the early burst model as the model expected of an adaptive radiation resulting from a key innovation that facilitates crossing an adaptive threshold and entering a new adaptive zone, followed by slower rates with the filling and subdivision of ecological niches. This view fits in with a recently proposed, and somewhat restrictive stance on key innovations. In the end, the authors found poor fit of venom toxins to the early burst model and better fit to pulsed models. This result implies that venom toxins do not just enable entering a new adaptive zone. Rather venom toxin continues to evolve with rapid pulsed episodes and to play a major role in the adaptations of snakes for capturing prey, perhaps enabling the coexistence of species, which supports higher standing diversity. In this respect, snake venom is not an anomaly, but rather is within a class of key innovations that promote further evolutionary change. Such adaptations are evolutionarily versatile in the sense of being highly evolvable and able to adapt quickly to changing conditions.

I feel that the authors could make a more streamlined case in their Introduction if they took a more open-ended view of key innovations from the onset and discussed the evolutionary models (early burst and pulsed) that are consistent with scenarios on how different kinds of key innovations might operate. Doing so changes the focus from whether venom is a key innovation, which ultimately depends on one's definition of the term, to how venom has evolved and influenced the evolutionary radiation of venomous snakes.

Thank you for this important feedback. We agree with the reviewer and thus have replaced the section about using gene expression data (which had its own shortcomings), with a section talking more about the different evolutionary models and their relevance in studies of key innovations, their limitations, and the use of pulsed models to account for the limitations.

We include the following:

Ln79-98: Evolutionary models are extensively used to study trait evolution, and have been used to model everything from body shape evolution to gene expression level evolution (Freckleton et al. 2003; Brawand et al. 2011). Therefore, it is not surprising that evolutionary models are also widely used to study key innovation. However, rarely does one model consistently explain the evolution of key innovations (or traits believed to be key innovations). Some traits are better explained by brownian motion (BM), others by Ornstein-Uhlenbeck (OU) models, some traits fit a single-peak OU model better, while others a multi-peak OU model, other traits fit neither BM or OU processes well (Price et al. 2010; Colombo et al. 2015; Poe et al. 2018; Burress et al. 2019; Arbour et al. 2019). Along with BM and OU models, it is also possible to model early burst (EB). An EB in speciation rate and trait evolution is believed to be the predominant pattern in adaptive radiation (11–13). While it is possible to model early burst evolution, evidence for it is rarely observed in comparative data (14). The often conflicting results between these models require cautious interpretation of features like evolutionary rates (Price et al. 2010). Perhaps one limitation of these models is using a Gaussian process to model continuous trait evolution. Evolutionary processes can result in changes that are too abrupt to be accounted for by a Gaussian process (Landis et al. 2013). Pulsed models offer one solution. Utilizing a Levy process, the pulsed models can account for abrupt shifts in the continuous character evolution, that conventional evolutionary models cannot easily explain (Landis et al. 2013). Using this approach Landis and Schraiber found that body size evolution is better represented by rare stochastic pulses of diversification than convention EB or multi-optima OU models (Landis and Schraiber 2017; Martin and Richards 2019). Therefore, examining traits using a pulsed model of evolution might reveal previously unresolved trends of evolution.

Specific Comments

Title

The title summarizes the main results of this study about the tempo and mode of toxin evolution in snakes, but it does not fully convey what the paper is really about, which is whether snake venom is a key innovation and how it has influenced the radiation of snakes. Perhaps the title ought to reflect the full scope of the paper.

We appreciate the reviewer's encouragement here. However, we decided to take a conservative stance and have the title reflect only our main results. Since we're trying to challenge a somewhat

strong belief in the field, introducing the challenge with a narrative build rather than with a strong claiming title would allow a gentle transition to the reader and not impose a bias from the off-set.

Abstract

Line 40. "are essential for prey capture" should perhaps be "facilitate prey capture". The latter wording is more active. Also, venom is essential only if a snake is restricted to a certain predation strategy (strike, follow, and wait) that requires it. Other snakes (boas and pythons) can also capture prey without venom, but with a different strategy and adaptations.

Agreed. We have modified the abstract.

Line 51. Change "influence" to "influences". Insert "in" after "process" and "that" after "way". Also, "magic traits" is an odd phrase that requires explanation or at least a reference. Therefore, it may be better used in the main text than here in the abstract where it can neither be explained nor a reference cited.

We have modified this line and removed 'magic traits'.

Introduction

Line 60. "ecological speciation". Some authors (e.g., Mike Rosenzweig) have used this term to mean a kind of sympatric speciation, whereas others do not and treat it as a kind of allopatric speciation. The authors should probably explain what they themselves mean by using the term.

Done.

Ln72-73: Ecological speciation, i.e. speciation driven by differences in ecology, is considered the primary mode by which adaptive radiation can take place, and as various traits produce specific differences in ecology, certain traits are more strongly associated with the radiation process than others (3,11)

Lines 67-70. "Rather, the role of key innovations should be restricted to providing entry into novel ecological niches or adaptive zones, and studies should aim to identify specific shifts in tempo and mode of phenotypic evolution of the assumed key innovation (9,10)." If the authors wish to adopt this restricted definition of key innovation, or more precisely criteria for recognizing a key innovation, that is fine. However, if adopting this definition leads them to use the Early Burst model as the sine qua non of a key innovation, as they seem to do in the following paragraph, then they undermine their own argument. Ultimately, the authors find that the evolution of venom toxins in snakes is better explained by Pulsed models than by an Early Burst model. That does not surprise me because venom toxin seems like a evolutionarily versatile system as the authors describe it, a lot like the pharyngeal jaws of cichlid fishes in sense of facilitating further, rapid evolutionary change. Finding pulsed rate of evolution is what I would expect of a key innovation that promotes further evolutionary change.

We thank the reviewer for this feedback. As mentioned above, we have streamlined our argument more by focussing on evolution of the snake venom phenotype itself rather than by estimating shifts in evolutionary rates and testing various trait evolution models.

The next thing that I would want to know is what are the tradeoffs. Are there any costs to having venom? How about certain combinations of toxins? I would advise the authors to frame their argument about venom as a key innovation in comparison to other innovations that are versatile and highly evolvable. Such adaptations tend to be those with low costs to the organisms, often through relaxed trade-offs among functions.

This is an important point that we actually decided to include in the conclusion. Doing so reinstated our motivation of using snake venom to study the evolution of key innovations.

Ln382-389: Snakes usually need to produce large amounts of venom, and determining if venom is more costly compared to other offensive (or defensive strategies) is difficult, as it requires prey-handling experiments, taxon specific toxicity testing, etc which are both complicated, and difficult to implement (Jackson et al. 2019). However, considering the several ways snakes can modulate venom output (eg venom metering, secretions with reduced protein content etc), venom might actually be an effective way of procuring energy rich meals (by subduing large prey) making it a particularly cost-effective innovation (Schendel et al. 2019; Jackson et al. 2019).

Line 85. "rarely observed in data (14)." The authors should clarify what kind of data. Comparative data in extant clades, I should think. In the fossil record, the early burst pattern is more commonplace but by no means universal.

Ln88-89: Have modified the text.

Lines 85-99 "This lack of empirical support might be because studies are overlooking the components of traits that actually produce phenotypic change; gene expression variation" and continuing to the end of the paragraph. I did not find this particular argument to be a convincing explanation for why few examples of early burst adaptive radiations occur among extant species. I do understand the authors' main point that rates of evolution of genes and phenotypes may differ, and I understand the authors' need to motivate their study around a specific question and a hypothesis. However, in adaptive evolution, genes are only going to evolve as rapidly as the phenotypes to which they correspond are sorted by natural selection. One way in which genes might evolve faster is in drift among selectively neutral variants. But probably would not be the case here. Alternatively, and perhaps more of a problem, is in cases of highly developmentally canalized systems in which the same phenotype can develop from more than one, and perhaps multiple different gene expression patterns. This situation may be commonplace and could lead to higher rates of evolution in genes than in phenotype.

We agree with the reviewer here and indeed feel our original motivation might have been a bit forced. As the reviewer showed interest in the study of the venom trait itself, we have decided to shift focus exclusively to understanding the evolutionary dynamics of gene expression in snake venom toxins. We have modified the introduction as mentioned previously (Ln79-98).

Lines 100-127. The problem of gene expression-phenotype mapping in complex traits versus in snake venom. The authors do make a good case here for the special status of venom because of its simplicity, in which different levels of gene expression result directly in different proportions among toxins in the venom “cocktail”. In other words, in snake venom one need not consider gene networks, morphogenetic pathways, and the like. The system sounds ideal. However, several thoughts occurred to me as I read this passage. What are the observed performance differences among toxin cocktails? What specific associations exist between toxin mixtures and either prey or environments? The idea that toxins do not interact with other is presented rather definitively and with citation of just one supporting reference. How was this inference made?

Thank you for the comment. In an attempt to answer the reviewer’s comments we have included the following in the introduction:

Ln116-122: Venom toxins can have both agonistic and antagonistic interactions between other toxin components, but how they influence other traits outside the venom system is unclear. On one hand venoms are integrated systems with different toxins acting in concert to immobilise prey (Reeks et al. 2015). On the other hand, whether this mode of action introduces an evolutionary constraint is less clear, since there is little phylogenetic covariance between components, and gene-environment constraints appear to act on individual loci, independent of co-expression patterns between toxin genes (Barua and Mikheyev 2019; Zancolli et al. 2019).

Although the authors present toxins within venom as isolated adaptations without interactions, venom requires a delivery system (glands and fangs) as they later mention briefly, a specific hunting strategy (behavior and supporting morphology, including sensations), and either immunity from the effects of one’s own toxins, some barrier sequestering toxins, or at least storage of toxins in an inactive state until deployed. How is all this potential complexity managed? One could imagine far-reaching consequences of having venom or of variation among venom cocktails on the phenotypes of snakes. Is venom really such an isolated system?

The reviewer is right in saying that the venom system has individual components which require a certain degree of coordination to function effectively. We didn’t mean to undermine these components, but rather stress on the fact that there is no quantitative evidence of how these components interact. Whatever associations exist, are for the most part, speculation. But as the reviewer mentions, there could well be numerous interactions which could be tested. Perhaps in a future publication. For the current study, we include the following which acknowledges the above mentioned interactions.

Ln350-354: The venom system comprises venom toxins, venom glands, fangs, and muscle architecture responsible for delivering the venom into the prey. Although the toxins are directly involved in prey immobilization, evolution of the other non-toxin components are essential for development of the venom system as a whole. Numerous examples exist of toxin recruitments coinciding with development of various morphological features like high-pressure venom delivery, and certain hunting strategies like ambush feeding (Fry et al. 2008). However, any modifications to enhance prey procurement would be restricted to the venom system, and unlikely to affect changes in other parts of the animal (33).

Lines 133-135. "If snake venoms are key innovations we would expect to see shifts in their evolutionary rates in response to changes in optima." How are optima identified? Ideally, this would be by reference to evidence external to the snakes themselves, such as the timing of environmental changes from the paleoclimatic record, biogeographic dispersal events into new regions, or time of contact with novel prey. There is a certain circularity to inferring the existence of optima from the evolutionary rate dynamics of snake venom alone. Is there external evidence too?

Thank you for this feedback. We agree that the mention of optima in the introduction does seem a bit abrupt and out of place. For that reason we have removed it from the introduction, and have dedicated a paragraph in the discussion to focus on the type of optima and relevant evidence for the same:

Ln284-291: The high variability in the snake venom phenotype is likely due to the presence of various optima. A previous study showed that distribution of toxin families on the macroevolutionary scale can be explained by the presence of convergent phylogenetic optima (27). Furthermore, the effect of various environmental factors like temperature and longitudinal climatic gradient influence venom variation, hinting at the occurrence of optima that maintain disparate, locally adaptive venom complexes (28). Therefore, the shifts in phenotypic macroevolutionary rates are likely due to shifts between these optima.

Results

Lines 232-235. "Jump models have highest weighted AIC scores and are a better fit to snake b venom gene expression data as compared to conventional incremental models of evolution (Table 1; electronic supplementary material)." Are the jump models grouped together in Table 1 under the heading "Pulsed"? If yes, then please state so.

We have modified the table legend to reflect this. Thank you.

In general, yes, the "Pulsed" models all fit better than the random (BM and OU) and early burst (EB) models. Do the pulsed models fit significantly better? Also, can the "pulsed" models

distinguish oscillation between stasis and random walk, on the one hand, from sporadic pulses of directional change, on the other hand? In order to relate these models to those traditionally used to study evolutionary rates in paleontology (stasis, random walk, and directional evolution), it would be beneficial if the authors would explain the correspondences between these models and traditional ones.

In the methods we describe how the pulvR models different types of pulse processes: The Lévy process is represented mathematically using the Lévy-Khinchine representation, where one can compute variance of trait change along a branch of length t . We model stasis followed by rapid adaptation using a compound Poisson process. This is the Jump Normal (JN) process, which assumes jump sizes are drawn from a normal distribution. The other pulsed evolution model is Normal Inverse Gaussian (NIG) which uses an infinitely active Lévy process to model constant rapid adaptation.

We believe the JN is analogous to periods of stasis and random walk, while NIG is similar to sporadic pulses, but much more rapid. We are unsure whether there is any directionality per se. The pulsed models test for dynamics which are very different from conventional models. For example, it can test the effect of fluctuating population size on quantitative trait evolution, which would be impossible in conventional models (Landis et al. 2013). Furthermore, the package estimates BM and OU processes in the standard framework (by rescaling branch lengths as a function of model parameters) and then allows models comparison with pulsed models, to determine which model best suits the given data. In both (Landis and Schraiber 2017) and (Landis et al. 2013), the authors use simulation to check the effect of model biases, and whether parameter estimates differ significantly from that of a BM process. In both instances the authors found that the model parameters are consistently significant and have low model bias.

The intricacies and nuances of the Levy process and how it models evolution are indeed fascinating, but beyond the scope of our discussion. We mentioned in the supplementary (which has all the models estimates), that readers interested can refer to the appendix of (Landis and Schraiber 2017), which is much more detailed.

Lines 273-281. Optima, prey diversity, and environment. So far, the existence of optima has only been inferred in the authors' data from the good fit of the data to pulsed (Lévy process) models that assume optima. The authors could make a stronger argument if they could be more specific about what the optima are and how venom has evolved specifically to allow snakes to adapt to these shifting optima.

We have clarified this in the discussion (Ln284-291).

If specific associations are lacking, then at the very least they could explain how prey diversity and environment could in theory co-define optima as the authors claim. For example, what toxin combinations are better suited to a diverse range of prey, and which are better suited to one or a few prey types? Do venomous snakes conform to the specialist/generalist dichotomy? Are there

tradeoffs between toxin efficacy on one kind of prey and the number of prey types on which it can be used? How does the environment factor in? Through general levels of metabolic and behavioral activity expected of an ectotherm? Or in more specific ways through prey availability, overall or seasonally? Right now, these optima are just a little too vague to be totally satisfying.

The points the reviewer mentions are important and definitely worth exploring. However, they are beyond the scope of the current paper which deals more with phenotypic rate dynamics, and not the effect of phenotypic variation in venom. The manuscript does include a section with reference to other studies that deal with this explicitly:

Ln124-127: Changes in expression levels of individual toxins alter their abundance in the venom, thereby influencing venom efficacy (39–41). This alteration in venom efficacy impacts the feeding ecology of snakes, which in turn determines how snakes adapt and colonize new niches (42,43).

As we have explained the origin of the optima in previous sections, we hope this satisfies the reviewer's queries.

Lines 346-349. "Nearly all studies of adaptive [change to "adaptation"] focus on traits and processes in extant species, [and] this is a major disadvantage since there is no way of representing extinct taxa and thus no way of determining whether a clade with specific innovations was more species[.]rich in the past (77,78)." Well said, and very refreshing to see this statement here. Please note minor edits in brackets.

Thank you for the comment. We have made the appropriate edits. (Ln370-371).

We thank the reviewer for their interest in our work. We hope the changes made are satisfactory, and welcome any new suggestion they might have.